# TEST-TIME ADAPTATION FOR BETTER ADVERSARIAL ROBUSTNESS

## ABSTRACT

Standard adversarial training and its variants have been widely adopted in practice to achieve robustness against adversarial attacks. However, we show in this work that such an approach does not necessarily achieve near optimal generalization performance on test samples. Specifically, it is shown that under suitable assumptions, Bayesian optimal robust estimator requires test-time adaptation, and such adaptation can lead to significant performance boost over standard adversarial training. Motivated by this observation, we propose a practically easy-to-implement method to improve the generalization performance of adversarially-trained networks via an additional self-supervised test-time adaptation step. We further employ a meta adversarial training method to find a good starting point for test-time adaptation, which incorporates the test-time adaptation procedure into the training phase and it strengthens the correlation between the pre-text tasks in self-supervised learning and the original classification task. Extensive empirical experiments on CIFAR10, STL10 and Tiny ImageNet using several different self-supervised tasks show that our method consistently improves the robust accuracy of standard adversarial training under different white-box and black-box attack strategies.

## 1 INTRODUCTION

*Adversarial Training* (AT) (Madry et al., 2018) and its variants (Wang et al., 2019; Zhang et al., 2019) are currently recognized as the most effective defense mechanism against adversarial attacks. However, AT generalizes poorly; the robust accuracy gap between the training and test set in AT is much larger than the training-test gap in standard training of deep networks (Neyshabur et al., 2017; Zhang et al., 2017). Unfortunately, classical techniques to overcome overfitting in standard training, including regularization and data augmentation, only have little effect in AT (Rice et al., 2020).

Theoretically, as will be shown in Section 3, the loss objective of AT does not achieve optimal robustness. Instead, under suitable assumptions, the Bayesian optimal robust estimator, which represents the statistical optimal model that can be obtained from training data, requires test-time adaptation. Compared with the fixed restricted Bayesian robust estimators, the test-time adapted estimators largely improve the robustness. Therefore, we should perform the test-time adaptation for each test input to boost the robustness.

To this end, we propose to fine-tune the model parameters for each test mini-batch. Since the labels of the test images are not available, we exploit self-supervision, which is widely used in the standard training of networks (Chen et al., 2020b; Gidaris et al., 2018; He et al., 2020). Fine-tuning the self-supervised tasks has a high gradient correlation with fine-tuning the classification task so that it forms a substitute of fine-tuning the classification loss at the inference time. Thus, we expect minimizing this self-supervised loss function yields better generalization on the test set.

To make our test-time adaptation strategy effective, we need to search for a good starting point that achieves good robust accuracy after fine-tuning. As will be shown in our experiments, AT itself does *not* provide the optimal starting point. We therefore formulate the search for such start point as a bilevel optimization problem. Specifically, we introduce a *Meta Adversarial Training* (MAT) strategy dedicated to our self-supervised fine-tuning inspired by the model-agnostic meta-learning (MAML) framework (Finn et al., 2017). To this end, we treat the classification of each batch of adversarial images as one task and minimize the corresponding classification error of the fine-tuned

network. MAT strengthens the correlation between the self-supervised and classification tasks so that self-supervised test-time adaptation can further improve robust accuracy.

In order to reliably evaluate our method, we follow the suggestions of (Tramer et al., 2020) and design an adaptive attack that is fully aware of the test-time adaptation. Using rotation and vertical flip as the self-supervised tasks, we empirically demonstrate the effectiveness of our method on the commonly used CIFAR10 (Krizhevsky et al., 2009), STL10 (Coates et al., 2011) and Tiny ImageNet (Le & Yang, 2015) datasets under both standard (Andriushchenko et al., 2020; Croce & Hein, 2020a; Madry et al., 2018) and adaptive attacks in both white-box and black-box attacks. The experiments evidence that our method consistently improves the robust accuracy under all attacks. Our contributions can be summarized as follows:

1. We show that the estimators should be test-time adapted in order to achieve the Bayesian optimal adversarial robustness, even for simple models like linear models. And the test-time adaptation largely improves the robustness compared with optimal restricted estimators.

2. We introduce the framework of self-supervised test-time fine-tuning for adversarially-trained networks, showing that it improves the robust accuracy of the test data.

3. We propose a meta adversarial training strategy based on the MAML framework to find a good starting point and strengthen the correlation between the self-supervised and classification tasks.

4. The experiments show that our approach is valid on diverse attack strategies, including an adaptive attack that is fully aware of our test-time adaptation, in both white-box and black-box attacks.

## 2 RELATED WORK

**Adversarial Training.** In recent years, many approaches have been proposed to defend networks against adversarial attacks (Guo et al., 2018; Liao et al., 2018; Song et al., 2018). Among them, *Adversarial Training* (AT) (Madry et al., 2018) stands out as one of the most robust and popular methods, even under various strong attacks (Athalye et al., 2018; Croce & Hein, 2020a). AT optimizes the loss of adversarial examples to find parameters that are robust to adversarial attacks. Several variants of AT (Wang et al., 2019; Zhang et al., 2019) also achieved and similar performance to AT (Rice et al., 2020). One important problem that limits the robust accuracy of AT is overfitting. Compared with training on clean images, the gap of robust accuracy between the training and test set is much larger in AT (Rice et al., 2020). Moreover, traditional techniques to prevent overfitting, such as regularization and data augmentation, have little effect. Recently, some methods have attempted to flatten the weight loss landscape to improve the generalization of AT. In particular, *Adversarial Weight Perturbation* (AWP) (Wu et al., 2020) achieves this by designing a double-perturbation mechanism that adversarially perturbs both inputs and weights. In addition, learning-based smoothing can flatten the landscape and improve the performance (Chen et al., 2021b).

**Self-supervised Learning.** In the context of non-adversarial training, many self-supervised strategies have been proposed, such as rotation prediction (Gidaris et al., 2018), region/component filling (Criminisi et al., 2004), patch-base spatial composition prediction (Trinh et al., 2019) and contrastive learning (Chen et al., 2020b; He et al., 2020). While self-supervision has also been employed in AT (Chen et al., 2020a; Kim et al., 2020; Yang & Vondrick, 2020; Hendrycks et al., 2019), their methods only use self-supervised learning at training time to regularize the parameters and improve the robust accuracy. By contrast, we propose to perform self-supervised fine-tuning at test time, which we demonstrate to significantly improve the robust accuracy on test images. As will be shown in the experiments, the self-supervised test-time adaptation has larger and complementary improvements over the training time self-supervision.

**Test-time Adaption.** Test-time adaptation has been used in various fields, such as image super-resolution (Shocher et al., 2018) and domain adaption (Sun et al., 2020; Wang et al., 2021). While our work is thus closely related to *Test-Time Training* (TTT) in (Sun et al., 2020), we target a significantly different scenario. TTT assumes that all test samples have been subject to the same distribution shift compared with the training data. As a consequence, it *incrementally* updates the model parameters when receiving new test images. By contrast, in our scenario, there is no systematic distribution shift, and it is therefore more effective to fine-tune the parameters of the *original* model for every new test mini-batch. This motivates our MAT strategy, which searches for the initial model parameters that can be effectively fine-tuned in a self-supervised manner.

## 3 THEORY OF TEST-TIME ADAPTATION

In this section, we study the relationship between the test-time adaptation and Bayesian optimal robustness, which represents the optimal robustness that can be achieved from the training data, showing that the test-time adaptation can extend the function classes and improve the robustness of the model.

**Definition 3.1.** *For a model $F(\mathbf{x})$ with $\ell_2$ adversarial constraint $\|\mathbf{x}^\star - \mathbf{x}\| < \varepsilon$, we define its natural risk and adversarial risk as at point $\mathbf{x}$*

$$R_{\mathbf{x}}^{nat}(F) = (F(\mathbf{x}) - \mathbb{E}[y|\mathbf{x}])^2, \quad R_{\mathbf{x}}^{adv}(F) = R_{\mathbf{x}}^{nat}(F) + \max_{\|\mathbf{x}^\star - \mathbf{x}\| < \varepsilon} (F(\mathbf{x}^\star) - F(\mathbf{x}))^2$$

**Remarks.** We use the MSE loss to define the natural risk $R_{\mathbf{x}}^{\mathrm{nat}}(F)$ and adversarial $R_{\mathbf{x}}^{\mathrm{adv}}(F)$ at point $\mathbf{x}$. Similar to TRADES (Zhang et al., 2019) The adversarial risk is defined as the sum of natural risk and the loss changes under adversarial attack, and it can be bounded by the maximum MSE loss within the adversarial budget

$$\mathbb{E}_{\mathbf{x}} R_{\mathbf{x}}^{\mathrm{adv}}(F) \leq \mathbb{E}_{\mathbf{x}} \max_{\|\mathbf{x}^\star - \mathbf{x}\| \leq \varepsilon} (F(\mathbf{x}^\star) - \mathbb{E}[y|\mathbf{x}])^2 \leq 2\mathbb{E}_{\mathbf{x}} R_{\mathbf{x}}^{\mathrm{adv}}(F).$$

Therefore, for the adversarial input $\mathbf{x}^\star$ with $\|\mathbf{x}^\star - \mathbf{x}\| < \varepsilon$, small $R_{\mathbf{x}}^{\mathrm{adv}}(F)$ guarantee small test error on $\mathbf{x}^\star$. Small $R_{\mathbf{x}}^{\mathrm{nat}}(F)$ and $R_{\mathbf{x}}^{\mathrm{adv}}(F)$ represents good clean performance and high adversarial robustness respectively

In the following definitions, we define three algorithms to obtain adversarially robust functions and compare their adversarial risks.

**Definition 3.2** (Adversarial Training with TRADES)**.** *We define $\hat{F}_{AT}$ as*

$$\hat{F}_{AT} = \arg\min_{\hat{F}} \frac{1}{n} \sum_{i=1}^{n} \left[ (y_i - \hat{F}(\mathbf{x}_i))^2 + \max_{\|\mathbf{x}_i^\star - \mathbf{x}_i\| < \varepsilon} (\hat{F}(\mathbf{x}_i^\star) - \hat{F}(\mathbf{x}_i))^2 \right],$$

*where $\mathbf{x}_i$ represents the $i$-th clean training data and $y_i$ represents the clean training label.*

**Remark.** Empirically, adversarial training is a very popular method to achieve robustness. It minimizes the adversarial risk on the training data. Then we consider the Bayesian optimal robustness. Let $\mathcal{F}$ represent a function class. We assume the response is generated by $y = F_*(\mathbf{x}) + \xi$ with prior distribution $F_* \in \mathcal{F} \sim \mathbb{P}_F$ and $\xi \sim \mathcal{P}_\xi$. Denote $\mathbf{X} \in \mathbb{R}^{n \times d}, \mathbf{Y} \in \mathbb{R}^n$ as the training data and training response generated by $y_i = F_*(\mathbf{x}_i) + \xi$. For problems like Bayesian linear regression (Bishop & Nasrabadi, 2006), function $\hat{F} \in \mathcal{F}$ is able to achieve the Bayesian optimal natural risk $\mathbb{E}_{\mathbf{X}, \mathbf{Y}, y}(\hat{F}(\mathbf{x}) - y)^2$. However, the function class $\mathcal{F}$ is not enough to achieve the Bayesian optimal adversarial risk. The adversarial risk depends on the local Lipschitz of function $\hat{F}$, in order to better trade-off between the Lipschitz and natural risk of the function $F$, much more complicated function classes than $\mathcal{F}$ are needed to achieve the optimal adversarial robustness. We defined the two Bayesian functions $F_{\mathrm{RB}}$ and $F_{\mathrm{AB}}$ that minimize global adversarial risk and adversarial risk at the specific point $\mathbf{x}$. $F_{\mathrm{AB}}$ extends the function class beyond $\mathcal{F}$ and achieves better robustness.

**Definition 3.3** (Restricted Bayesian Robust Function $F_{RB}$)**.** *The restricted Bayesian robust function $F_{RB}$ minimizes the global adversarial risk inside the function class $\mathcal{F}$*

$$\min_{\hat{F} \in \mathcal{F}} \mathbb{E}_{\mathbf{x}} \mathbb{E}_{y, \mathbf{X}, \mathbf{Y}|\mathbf{x}} \left[ (\hat{F}(\mathbf{x}) - y)^2 + \max_{\|\mathbf{x}^\star - \mathbf{x}\| < \varepsilon} (\hat{F}(\mathbf{x}^\star) - \hat{F}(\mathbf{x}))^2 \right]$$

**Remark.** The Bayesian function represents the best robust function inside the function class $\mathcal{F}$ For any $F \in \mathcal{F}$, no training algorithms can achieve better average adversarial risk than $F_{\mathrm{RB}}$.

**Definition 3.4** (Adaptive Bayesian Robust Function $F_{AB}$)**.** *The adaptive Bayesian robust function $F_{AB}$ inside the function class $\mathcal{F}$ that minimizes the adversarial risk at point $\mathbf{x}$ is*

$$\min_{\hat{F} \in \mathcal{F}} \mathbb{E}_{y, \mathbf{X}, \mathbf{Y}|\mathbf{x}} \left[ (\hat{F}(\mathbf{x}) - y)^2 + \max_{\|\mathbf{x}^\star - \mathbf{x}\| < \varepsilon} (\hat{F}(\mathbf{x}^\star) - \hat{F}(\mathbf{x}))^2 \right]$$

**Remark.** Instead of minimizing the global average $R^{adv}_{\mathbf{x}}(F)$, $F_{AB}$ minimizes the adversarial risk **in the given input point** $\mathbf{x}$. The function depends on the input $\mathbf{x}$ so that the model extends the function class beyond $\mathcal{F}$. For different test inputs, we can use different functions to achieve the optimal adversarial risk. Therefore, we refer to $F_{AB}$ as the test-time adapted function.

In the following theorem, we show the difference between three functions in the model, where the test-adapted function $F_{AB}$ significantly improves the robustness.

**Theorem 3.1** (Linear Models). *We consider a linear function classes $\mathcal{F}^{Lin} = \{F^{Lin} | F^{Lin}(\mathbf{x}; \boldsymbol{\theta}) = \mathbf{x}^\top \boldsymbol{\theta}, \boldsymbol{\theta} \in \mathbb{R}^d\}$. The output $y$ is generated by $y = \mathbf{x}^\top \boldsymbol{\theta}_* + \xi$, where $\boldsymbol{\theta}_*$ is independent of $\mathbf{x}$ with $\boldsymbol{\theta}_* \sim \mathcal{N}(0, \tau^2 \mathbf{I})$, and the noise $\xi \sim \mathcal{N}(0, \sigma^2)$. Let $\mathbf{X} \in \mathbb{R}^{n \times d}, \mathbf{Y} \in \mathbb{R}^n$ denote the training data and the responses respectively. For linear model $F^{Lin}(\mathbf{x}; \boldsymbol{\theta})$, three estimators in Definition 3.2 to 3.4 are*

$$\widehat{\boldsymbol{\theta}}^{Lin}_{AT} = \mathbf{X}(\mathbf{X}^\top \mathbf{X} + n\varepsilon^2 \mathbf{I}_n)^{-1}\mathbf{Y}, \quad \widehat{\boldsymbol{\theta}}^{Lin}_{RB} = \frac{1}{\varepsilon^2 d + 1}\widehat{\boldsymbol{\theta}}^{Lin}_{nat}, \quad \widehat{\boldsymbol{\theta}}^{Lin}_{AB} = (\mathbf{x}\mathbf{x}^\top + \varepsilon^2 \mathbf{I}_d)^{-1}\mathbf{x}\mathbf{x}^\top \widehat{\boldsymbol{\theta}}^{Lin}_{nat},$$

*where $\widehat{\boldsymbol{\theta}}^{Lin}_{nat} = \mathbf{X}(\mathbf{X}^\top \mathbf{X} + \lambda_* \mathbf{I}_n)^{-1}\mathbf{Y}$, $\lambda_* = \sigma^2/\tau^2$. Furthermore, if each dimension of $\mathbf{x}$ is i.i.d. with $\mathbb{E}\mathbf{x} = 0$, $Cov(\mathbf{x}) = \mathbf{I}_d/d$ and $\mathbb{E}[\sqrt{d}x^i]^4 \leq M$ for some universal constant $M < \infty$, denoting $\Delta = (1 + c + \lambda_*)^2 - 4c$, then when $n, d \to \infty$ with $n/d = c \in (0, 1)$*

$$R^{adv}_{\mathbf{x}}(\widehat{\boldsymbol{\theta}}^{Lin}_{AT}) = \tau^2, \quad R^{adv}_{\mathbf{x}}(\widehat{\boldsymbol{\theta}}^{Lin}_{RB}) = \tau^2, \quad R^{adv}_{\mathbf{x}}(\widehat{\boldsymbol{\theta}}^{Lin}_{AB}) = \tau^2 \left(1 - \frac{1 + c + \lambda_* - \sqrt{\Delta}}{2(\varepsilon^2 + 1)}\right).$$

*And when $c \to 1$ with SNR$= \sigma^2/\tau^2 \leq 1$, $R^{adv}_{\mathbf{x}}(\widehat{\boldsymbol{\theta}}^{Lin}_{AB}) < R^{adv}_{\mathbf{x}}(\widehat{\boldsymbol{\theta}}^{Lin}_{RB})(1 - \frac{2}{3(\varepsilon^2+1)})$.*

**Remarks.** In this theorem, we provide the form of the three estimators and their adversarial risks. The gap of adversarial risk between $\widehat{\boldsymbol{\theta}}^{Lin}_{AT}$ and $\widehat{\boldsymbol{\theta}}^{Lin}_{RB}$ vanishes when $n, d \to \infty$. The estimator $\widehat{\boldsymbol{\theta}}^{Lin}_{RB}$ achieves the optimal robust risk among all linear models. However, for an arbitrary ratio $c = n/d$, $R^{adv}_{\mathbf{x}}(\widehat{\boldsymbol{\theta}}^{Lin}_{AB}) < R^{adv}_{\mathbf{x}}(\widehat{\boldsymbol{\theta}}^{Lin}_{RB})$, **indicating that adaptation to each test data $\mathbf{x}$ can improve the robustness of the model even when compared with the best linear model.**

Theorem 3.1 provides the optimal test-time adapted estimator in the linear function classes $\mathcal{F}^{Lin}$, which depends on the clean input $\mathbf{x}$. In Figure 1, we plot the adversarial risk of three estimators for different adversarial budgets, which **clearly shows that our adaptation can significantly increase the robustness.** When the input is corrupted with adversarial noise, the same form of the test-time adapted estimator also significantly improves the adversarial risk shown in the following theorem.

**Theorem 3.2** (Corrupted Input). *We assume the oracle parameter $\boldsymbol{\theta}_*$ is independent of $\mathbf{x}$ and has the prior distribution $\boldsymbol{\theta}_* \sim \mathcal{N}(0, \tau^2 \mathbf{I})$, and the noise $\xi \sim \mathcal{N}(0, \sigma^2)$. Furthermore, each dimension of $\mathbf{x}$ is i.i.d. with $\mathbb{E}\mathbf{x} = 0$, $Cov(\mathbf{x}) = \mathbf{I}_d/d$ and $\mathbb{E}[\sqrt{d}x^i]^4 \leq M$ for some universal constant $M < \infty$, then when $n, d \to \infty$ with $n/d = c \in (0, 1)$. Given corrupted input $\mathbf{x}^\star = \mathbf{x} + \varepsilon \widehat{\boldsymbol{\theta}}/\|\widehat{\boldsymbol{\theta}}\|$, with $\varepsilon < 1$, the adversarial risk of $\widehat{\boldsymbol{\theta}}^{Lin}_{AB,\star} = (\mathbf{x}^\star \mathbf{x}^{\star\top} + \varepsilon^2 \mathbf{I}_d)^{-1}\mathbf{x}^\star \mathbf{x}^{\star\top}\mathbf{X}(\mathbf{X}^\top \mathbf{X} + \lambda_* \mathbf{I}_n)^{-1}\mathbf{Y}$ is*

$$R^{adv}_{\mathbf{x}}(\widehat{\boldsymbol{\theta}}^{Lin}_{AB,\star}) = \tau^2 \left(1 - (1 - \varepsilon^2 + \frac{2\varepsilon^2}{(1+\varepsilon)^2})\frac{2c}{(1 + c + \lambda_* + \sqrt{\Delta})(\varepsilon^2/(1+\varepsilon)^2 + 1)}\right) < R^{adv}_{\mathbf{x}}(\widehat{\boldsymbol{\theta}}^{Lin}_{RB}).$$

**Remarks.** The theorem shows that when the given input is adversarial, the test-time adaptation can still lower the adversarial risk of the model as $R^{adv}_{\mathbf{x}}(\widehat{\boldsymbol{\theta}}^{Lin}_{AB}) < R^{adv}_{\mathbf{x}}(\widehat{\boldsymbol{\theta}}^{Lin}_{AB,\star}) < R^{adv}_{\mathbf{x}}(\widehat{\boldsymbol{\theta}}^{Lin}_{RB})$.

**In the statistical Bayesian model, we show that the test-time adaptation can extend the function classes and achieve the significantly lower adversarial risk than the fixed model.** In the practical non-Bayesian classification task, explicit calculation of the optimal model is difficult. Nevertheless, the test-time adaptation also helps to improve the robustness of the model. As will be shown in the following section, we perform the self-supervised test-time fine-tuning to adapt the model to each input, and largely improves the robust accuracy of the test-time adapted model.

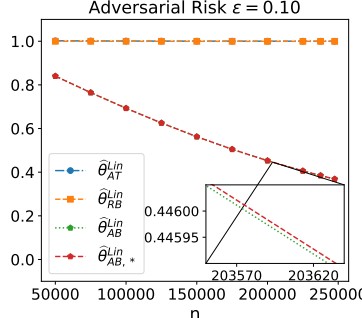

Figure 1: The comparison of $R^{adv}_{\mathbf{x}}$ for $\widehat{\boldsymbol{\theta}}^{Lin}_{AT}$, $\widehat{\boldsymbol{\theta}}^{Lin}_{RB}$, $\widehat{\boldsymbol{\theta}}^{Lin}_{AB}$, $\widehat{\boldsymbol{\theta}}^{Lin}_{AB,\star}$. We set $\tau^2 = 1$, $\sigma^2 = 0.2$ and $d = 250000$.

## 4 METHODOLOGY

We follow the traditional multitask learning formulation (Caruana, 1997) and consider a neural network with a backbone $\mathbf{z} = E(\mathbf{x}; \boldsymbol{\theta}_E)$ and $K+1$ heads. One head $f(\mathbf{z}; \boldsymbol{\theta}_f)$ outputs the classification result while the other $K$ heads $g_1(\mathbf{z}; \boldsymbol{\theta}_{g1}), ..., g_K(\mathbf{z}; \boldsymbol{\theta}_{gK})$ correspond to $K$ auxiliary self-supervised tasks. $\boldsymbol{\theta} = (\boldsymbol{\theta}_E, \boldsymbol{\theta}_f, \boldsymbol{\theta}_{g1}, \cdots, \boldsymbol{\theta}_{gK})$ encompasses all trainable parameters, and we further define

$$F = f \circ E; \quad G_k = g_k \circ E, \; k = 1, 2, ..., K. \tag{1}$$

Furthermore, let $D = \{(\mathbf{x}_i, y_i)\}_{i=1}^n$ denote the training set, and $\widetilde{D} = \{(\widetilde{\mathbf{x}}_i, \widetilde{y}_i)\}_{i=1}^m$ be the test set. For further illustration, the labels of the test set are shown. However, they are unknown to the networks at test time. We denote the adversarial examples of $\mathbf{x}$ as $\mathbf{x}^\star$. It satisfies $\|\mathbf{x}^\star - \mathbf{x}\| \leq \varepsilon$, and $\varepsilon$ is the size of the adversarial budget. For any set $S$, we represent its average loss as

$$\mathcal{L}(S) = \frac{1}{|S|} \sum_{s_i \in S} \mathcal{L}(s_i) \tag{2}$$

where $|S|$ is the number of elements in $S$. The general classification loss, such as the cross-entropy, is denoted by $\mathcal{L}_{cls}$. We use the superscript "AT" to denote the adversarial training loss. For example,

$$\mathcal{L}_{cls}^{AT}(S) = \frac{1}{|S|} \sum_{\mathbf{x}_i, y_i \in S} \max_{\|\mathbf{x}_i^\star - \mathbf{x}_i\| \leq \varepsilon} \mathcal{L}_{cls}(F(\mathbf{x}_i^\star), y_i) \, . \tag{3}$$

### 4.1 SELF-SUPERVISED TEST-TIME FINE-TUNING

Our goal is to perform self-supervised learning on the test examples to mitigate the overfitting problem of AT and adapt the model for each data point. To this end, let us suppose that an adversarially-trained network with parameters $\boldsymbol{\theta}^0$ receives a mini-batch of $b$ adversarial test examples $\widetilde{B}^\star = \{(\widetilde{\mathbf{x}}_1^\star, \widetilde{y}_1), \cdots, (\widetilde{\mathbf{x}}_b^\star, \widetilde{y}_b)\}$ , As the labels $\{\widetilde{y}_i\}_{i=1}^b$ are not available, we propose to fine-tune the backbone parameters $\boldsymbol{\theta}_E$ by optimizing the loss function

$$\mathcal{L}_{SS}(\widetilde{B}^\star) = \frac{1}{b} \sum_{k=1}^K C_k \sum_{i=1}^b \mathcal{L}_{SS,k}(G_k(\widetilde{\mathbf{x}}_i^\star); \boldsymbol{\theta}_E, \boldsymbol{\theta}_{gk}) \, , \tag{4}$$

which encompasses $K$ self-supervised tasks. Here, $\mathcal{L}_{SS,k}$ represents the loss function of the $k$-th task and $\{C_k\}_{k=1}^K$ are the weights balancing the contribution of each task. In our experiments, the $\mathcal{L}_{SS,K}$ is the cross-entropy loss to predict rotation and vertical flip.

The number of images $b$ may vary from 1 to $m$. $b = 1$ corresponds to the online setting, where only one adversarial image is available at a time, and the backbone parameters $\boldsymbol{\theta}_E$ are adapted to every new image. The online setting is the most practical one, as it does not make any assumptions about the number of adversarial test images the network receives. By contrast, $b = m$ corresponds to the offline setting, where all adversarial test examples are available at once. It is similar to transductive learning (Gammerman et al., 1998; Vapnik, 2013). Note that our online setting differs from the online test-time training described in TTT (Sun et al., 2020); we do not incrementally update the network parameters as new samples come, but instead initialize fine-tuning from the same starting point $\boldsymbol{\theta}^0$ for each new test image.

Eqn (4) encourages $\boldsymbol{\theta}_E$ to update in favor of the self-supervised tasks. However, as the classification head $f$ was only optimized for the old backbone $E(\cdot; \boldsymbol{\theta}_E^0)$, it will typically be ill-adapted to the new parameters $\boldsymbol{\theta}_E^*$, resulting in a degraded robust accuracy. Furthermore, for a small $b$, the model tends to overfit to the test data, reducing $\mathcal{L}_{SS}$ to 0 but extracting features that are only useful for the self-supervised tasks. To overcome these problems, we add an additional loss function acting on the training data that both regularizes the backbone $E$ and optimizes the classification head $f$ so that $f$ remains adapted to the fine-tuned backbone $E(\cdot; \boldsymbol{\theta}_E^*)$. Specifically, let $B \subset D$ denote a subset of the training set. We then add the regularizer

$$\mathcal{L}_R(B) = \mathcal{L}_{cls}^{AT}(B) = \frac{1}{|B|} \sum_{\mathbf{x}_i, y_i \in B} \max_{\|\mathbf{x}_i^\star - \mathbf{x}_i\| \leq \varepsilon} \mathcal{L}_{cls}(F(\mathbf{x}_i^\star), y_i) \tag{5}$$

to the fine-tuning process. In short, Eqn (5) evaluates the AT loss on the training set to fine-tune the parameters $\boldsymbol{\theta}_f$ of the classification head. It also forces the backbone $E$ to extract features that can

be used to make correct predictions, i.e., to prevent $\boldsymbol{\theta}_E$ from being misled by $\mathcal{L}_{SS}$ when $b$ is small. Combining Eqn (4) and Eqn (5), our final test-time adaptation loss is

$$\mathcal{L}_{test}(\widetilde{B}^\star, B) = \mathcal{L}_{SS}(\widetilde{B}^\star) + C\mathcal{L}_R(B) \tag{6}$$

where $C$ sets the influence of $\mathcal{L}_R$. The algorithms that describe our test-time self-supervised learning are deferred to Appendix D. As SGD is more efficient for larger amount of data, we use SGD to optimize $\boldsymbol{\theta}$ when $b$ is large (*e.g.* offline setting).

## 4.2 META ADVERSARIAL TRAINING

To make the best out of optimizing $\mathcal{L}_{test}$ at test time, we should find a suitable starting point $\boldsymbol{\theta}^0$, i.e., a starting point such that test-time self-supervised learning yields better robust accuracy. We translate this into a meta learning scheme, which entails a bilevel optimization problem.

Specifically, we divide the training data into $s$ small exclusive subsets $D = \cup_{j=1}^s B_j$ and let $B_j^\star$ to be adversaries of $B_j$. We then formulate meta adversarial learning as the bilevel minimization of

$$\mathcal{L}_{meta}(D;\boldsymbol{\theta}) = \frac{1}{s} \sum_{B_j \subset D} \mathcal{L}_{cls}^{AT}(B_j; \boldsymbol{\theta}_j^*(\boldsymbol{\theta})), \text{ where } \boldsymbol{\theta}_j^* = \arg\min_{\boldsymbol{\theta}} \mathcal{L}_{SS}(B_j^\star; \boldsymbol{\theta}) , \tag{7}$$

where $\mathcal{L}_{SS}$ is the self-supervised loss function defined in Eqn (4) and $\mathcal{L}_{cls}^{AT}$ is the loss function of AT defined in Eqn (3). As bilevel optimization is time-consuming, following MAML (Finn et al., 2017), we use a single gradient step of the current model parameters $\boldsymbol{\theta}$ to approximate $\boldsymbol{\theta}_j^*$.

$$\boldsymbol{\theta}_j^* \approx \boldsymbol{\theta} - \alpha\nabla_{\boldsymbol{\theta}}\mathcal{L}_{SS}(B_j^\star; \boldsymbol{\theta}) . \tag{8}$$

In essence, this Meta Adversarial Training (MAT) scheme searches for a starting point such that fine-tuning with $\mathcal{L}_{SS}$ will lead to good robust accuracy. If this holds for all training subsets, then we can expect the robust accuracy after fine-tuning at test time also to increase. Note that, because the meta learning objective of Eqn (7) already accounts for classification accuracy, the regularization by $\mathcal{L}_R$ is not needed during meta adversarial learning.

**Accelerating Training.** To compute the gradient $\nabla_{\boldsymbol{\theta}}\mathcal{L}_{meta}(D;\boldsymbol{\theta})$, we need to calculate the time-consuming second order derivatives $-\alpha\nabla_{\boldsymbol{\theta}}^2\mathcal{L}_{SS}(B_j^\star; \boldsymbol{\theta})\nabla_{\boldsymbol{\theta}_j^*}\mathcal{L}_{cls}^{AT}(B_j; \boldsymbol{\theta}_j^*)$ . Considering that AT is already much slower than standard training (Shafahi et al., 2019), we cannot afford another significant training overhead. Fortunately, as shown in (Finn et al., 2017), second order derivatives have little influence on the performance of MAML. We therefore ignore them and take the gradient to be

$$\nabla_{\boldsymbol{\theta}}\mathcal{L}_{meta}(D;\boldsymbol{\theta}) \approx \frac{1}{s} \sum_{B_j \subset D} \nabla_{\boldsymbol{\theta}_j^*}\mathcal{L}_{cls}^{AT}(B_j; \boldsymbol{\theta}_j^*) . \tag{9}$$

However, by ignoring the second order gradient, only the parameters on the forward path of the classifier $F$, *i.e.*, $\boldsymbol{\theta}_E$ and $\boldsymbol{\theta}_f$, will be updated. In other words, optimizing Eqn (7) in this fashion will not update $\{\boldsymbol{\theta}_{gk}\}_{k=1}^K$. To nonetheless encourage each self-supervised head $G_k$ to output the correct prediction, we incorporate an additional loss function encoding the self-supervised tasks,

$$\mathcal{L}_{SS}^{AT}(D) = \sum_k C_k\mathcal{L}_{SS,k}^{AT}(D) = \sum_k \frac{C_k}{|D|} \sum_{\mathbf{x}_i \in D} \max_{\|\mathbf{x}_i^\star - \mathbf{x}_i\| \le \varepsilon} \mathcal{L}_{SS,k}(G_k(\mathbf{x}_i^\star)) . \tag{10}$$

Note that we use the adversarial version of $\mathcal{L}_{SS}$ to provide robustness to the self-supervised tasks, which, as shown in (Chen et al., 2020a; Hendrycks et al., 2019; Yang & Vondrick, 2020), is beneficial for the classifier. The final meta adversarial learning objective therefore is

$$\mathcal{L}_{train}(D) = \mathcal{L}_{meta}(D) + C'\mathcal{L}_{SS}^{AT}(D) \tag{11}$$

where $C'$ balances the two losses. Algorithm 1 shows the complete MAT algorithm.

## 5 EXPERIMENTS

**Experimental Settings.** Following previous works (Cui et al., 2020; Huang et al., 2020), we consider $\ell_\infty$-norm attacks with an adversarial budget $\varepsilon = 0.031 (\approx 8/255)$. We evaluate our method

---

**Algorithm 1** Meta Adversarial Training

---

**Input:** Training set $D$; Learning rate $\alpha, \beta$; Iterations $T$; Weights $C_k$ and $C'$
**Output:** Starting parameters $\boldsymbol{\theta}_0$ for the test-time fine-tuning
1: **for** $t = 1$ to $T$ **do**
2:     Sample $q$ exclusive batches of training images $B_1, B_2, \cdots, B_q \subset D$
3:     Using PGD to find the adversaries $B_j^\star$: $\mathbf{x}_{j,i}^\star = \arg\max_{\|\mathbf{x}_{j,i}^\star - \mathbf{x}_{j,i}\| \leq \varepsilon} \mathcal{L}_{cls}(F(\mathbf{x}_{j,i}^\star), y_{j,i})$
4:     **for** batches $B_1, B_2, \cdots, B_q$ **do**
5:         $\boldsymbol{\theta}_j^* = \boldsymbol{\theta} - \alpha\nabla_{\boldsymbol{\theta}}\mathcal{L}_{SS}(B_j^\star; \boldsymbol{\theta})$
6:         $l_{meta,j} = \mathcal{L}_{cls}^{AT}(B_j; \boldsymbol{\theta}_j^*)$
7:     **end for**
8:     $\boldsymbol{\theta} = \boldsymbol{\theta} - \frac{\beta}{q}\sum_{B_j}\left[\nabla_{\boldsymbol{\theta}_j^*}l_{meta,j} + C'\nabla_{\boldsymbol{\theta}}\mathcal{L}_{SS}^{AT}(B_j; \boldsymbol{\theta})\right]$
9: **end for**
10: **return** Trained parameters $\boldsymbol{\theta}^0 = \boldsymbol{\theta}$

---

on three datasets: CIFAR10 (Krizhevsky et al., 2009), STL10 (Coates et al., 2011) and Tiny ImageNet (Le & Yang, 2015). We also use two different network architectures: WideResNet-34-10 (Zagoruyko & Komodakis, 2016) for CIFAR10, and ResNet18 (He et al., 2016) for STL10 and Tiny ImageNet. The hyperparameters are provided in the Appendix D.

**Self-Supervised Tasks.** In principle, any self-supervised tasks can be used for test-time fine-tuning, as long as they are positively correlated with the robust accuracy. However, for the test-time fine-tuning to remain efficient, we should not use too many self-supervised tasks. Furthermore, as we aim to support the fully online setting, where only one image is available at a time, we cannot incorporate a contrastive loss (Chen et al., 2020b; He et al., 2020; Kim et al., 2020) to $\mathcal{L}_{SS}$. In our experiments, we therefore use two self-supervised tasks that have been shown to be useful to improve the classification accuracy: *Rotation Prediction* and *Vertical Flip Prediction*.

**Attack Methods.** In the white-box attacks, the attacker knows every detail of the defense method. Therefore, we need to assume that the attacker is aware of our test-time adaptation method and will adjust its strategy for generating adversarial examples accordingly. Suppose that the attacker is fully aware of the hyperparameters for test-time adaptation. Then, finding adversaries $\widetilde{B}^\star$ of the clean subset $\widetilde{B}$ can be achieved by maximizing the adaptive loss

$$\widetilde{\mathbf{x}}_i^\star = \arg\max_{\|\widetilde{\mathbf{x}}_i^\star - \mathbf{x}_i\| \leq \varepsilon} \mathcal{L}_{attack}(F(\widetilde{\mathbf{x}}_i^\star), y; \boldsymbol{\theta}^T(\widetilde{B}^\star)) , \tag{12}$$

where $\mathcal{L}_{attack}$ refers to the general attack loss, such as the cross-entropy or the difference of logit ratio (DLR) (Croce & Hein, 2020a). We call this objective in Eqn (12) *adaptive attack*, which can be either performed in white-box or black-box attacks. We consider four common white-box and black-box attack methods: PGD-20 (Madry et al., 2018), AutoPGD (both cross-entropy and DLR loss) loss (Croce & Hein, 2020a), FAB (Croce & Hein, 2020b) and Square Attack (Andriushchenko et al., 2020). We apply both the standard and adaptive versions of these methods. Particularly, AutoPGD we use is a strong version that maximizes the loss function that continues when finding adversarial examples (Croce et al., 2022). More details are provided in the Appendix E.

**Baselines.** We compare our method with the following methods: 1) Regular AT, which uses $\mathcal{L}_{cls}^{AT}$ in Eqn (3). 2) Regular AT with an additional self-supervised loss, i.e., using $\mathcal{L}_{cls}^{AT} + C'\mathcal{L}_{SS}^{AT}$ for AT, where $\mathcal{L}_{SS}^{AT}$ is given in Eqn (10). This corresponds to the formulation of (Hendrycks et al., 2019). 3) MAT (Algorithm 1) without test-time fine-tuning.

## 5.1 ROBUST ACCURACY

**CIFAR10.** Table 1a shows the robust accuracy for different attacks and using two different tasks for fine-tuning. The adaptive attack is not applicable to models without fine-tuning. As we inject different self-supervised tasks into the AT stage, and as different self-supervised tasks may impact the robust accuracy differently (Chen et al., 2020a), the robust accuracy without fine-tuning still varies. The vertical flipping task yields better robust accuracy before fine-tuning but its improvement after fine-tuning is small. By contrast, rotation prediction achieves low robust accuracy before fine-tuning,

Table 1: Robust accuracy on CIFAR10, STL10 and Tiny ImageNet of the test-time fine-tuning on both the online and the offline settings. We use an $\ell_\infty$ budget $\varepsilon = 0.031$. FT stands for fine-tuning. We underline the accuracy of the strongest attack and highlight the highest accuracy among them.

(a) CIFAR10 with WideResNet-34-10.

| Tasks | Methods | Square Attack | | PGD-20 | | AutoPGD | | | FAB | | Worst |
|---|---|---|---|---|---|---|---|---|---|---|---|
| | | Standard | Adaptive | Standard | Adaptive | Standard | Adaptive | GMSA | Standard | Adaptive | |
| None | AT | 62.51% | - | 55.74% | - | 52.14% | - | - | 51.34% | - | 51.30% |
| Rotation | AT w/o FT | 63.54% | - | 56.64% | - | 52.57% | - | - | 51.87% | - | 51.85% |
| | MAT w/o FT | 63.96% | - | 57.35% | - | 53.09% | - | - | 53.09% | - | 53.04% |
| | Online FT | 65.52% | 65.85% | 59.52% | 59.50% | 57.93% | 56.96% | 57.60% | 75.58% | 77.69% | 56.62% |
| | Offline FT | 67.05% | 65.75% | 61.17% | 59.71% | 58.77% | 57.63% | - | 78.12% | 68.60% | **57.21%** |
| VFlip | AT w/o FT | 62.09% | - | 55.50% | - | 52.79% | - | - | 51.24% | - | 51.23% |
| | MAT w/o FT | 66.15% | - | 59.73% | - | 53.41% | - | - | 53.02% | - | 52.98% |
| | Online FT | 66.91% | 66.16% | 61.47% | 59.40% | 58.74% | 56.79% | 58.53% | 75.68% | 80.57% | 55.98% |
| | Offline FT | 67.23% | 65.60% | 61.82% | 59.69% | 59.26% | 58.06% | - | 75.60% | 72.24% | **57.01%** |
| Rotation + VFlip | AT w/o FT | 65.64% | - | 59.19% | - | 53.16% | - | - | 53.05% | - | 52.95% |
| | MAT w/o FT | 65.75% | - | 59.51% | - | 53.99% | - | - | 53.85% | - | 53.76% |
| | Online FT | 67.34% | 66.80% | 61.79% | 60.46% | 59.23% | 57.70% | 59.60% | 76.39% | 79.80% | 57.21% |
| | Offline FT | 68.50% | 66.05% | 62.87% | 60.54% | 60.25% | 58.26% | - | 76.89% | 71.58% | **57.88%** |

(b) STL10 with ResNet18.

| Tasks | Methods | Square Attack | | PGD-20 | | AutoPGD | | FAB | | Worst |
|---|---|---|---|---|---|---|---|---|---|---|
| | | Standard | Adaptive | Standard | Adaptive | Standard | Adaptive | Standard | Adaptive | |
| None | AT | 44.83% | - | 37.89% | - | 35.78% | - | 35.64% | - | 35.58% |
| Rotation + VFlip | AT w/o FT | 44.00% | - | 36.92% | - | 33.72% | - | 33.73% | - | 33.65% |
| | MAT w/o FT | 44.75% | - | 38.66% | - | 35.60% | - | 35.38% | - | 35.31% |
| | Online FT | 45.07% | 46.19% | 40.31% | 40.24% | 39.53% | 40.85% | 51.25% | 51.08% | 39.21% |
| | Offline FT | 47.86% | 48.03% | 45.21% | 43.33% | 43.78% | 43.20% | 58.49% | 54.13% | **42.57%** |

(c) Tiny ImageNet with ResNet18.

| Tasks | Methods | Square Attack | | PGD-20 | | AutoPGD | | FAB | | Worst |
|---|---|---|---|---|---|---|---|---|---|---|
| | | Standard | Adaptive | Standard | Adaptive | Standard | Adaptive | Standard | Adaptive | |
| None | AT | 28.5% | - | 20.6% | - | 17.5% | - | 17.2% | - | 17.2% |
| Rotation + VFlip | AT w/o FT | 29.5% | - | 22.2% | - | 17.1% | - | 16.7% | - | 16.7% |
| | MAT w/o FT | 29.3% | - | 23.1% | - | 16.9% | - | 16.8% | - | 16.7% |
| | Online FT | 30.2% | 30.2% | 24.0% | 23.2% | 18.9% | 18.1% | 33.7% | 31.6% | 17.7% |
| | Offline FT | 32.4% | 31.0% | 25.6% | 24.1% | 23.7% | 20.6% | 36.5% | 27.7% | **20.1%** |

Table 2: The statistics $\rho(\widetilde{\mathbf{x}}^\star)$ and two self-supervised tasks. The dataset is CIFAR10 and the network is WideResNet-34-10. Adversarial budget $\varepsilon = 0.031$

| Tasks | $\mathbb{E}(\rho(\widetilde{\mathbf{x}}^\star))$ | $P(\rho(\widetilde{\mathbf{x}}^\star) > 0)$ |
|---|---|---|
| Rotation | 0.15 | 68.51% |
| VFlip | 0.22 | 72.16% |

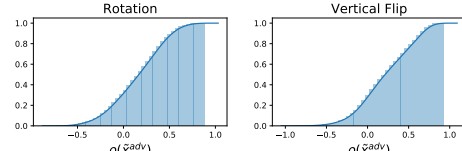

Figure 2: Empirical *cdf* of $\rho(\widetilde{\mathbf{x}}_i^\star)$ on CIFAR10 and WideResNet-34-10. Adversarial budget $\varepsilon = 0.031$

but its improvement after fine-tuning is the largest. Using both tasks together combines their effect and yields the highest overall accuracy after test-time adaptation. Note that our self-supervised test-time fine-tuning, together with meta adversarial learning, consistently improves the robust accuracy under different attack methods. Under the strongest adaptive AutoPGD, test-time fine-tuning using both tasks achieves a robust accuracy of 57.70%, significantly outperforming regular AT.

**STL10 and Tiny ImageNet.** As using both the rotation and vertical flip prediction led to the highest overall accuracy on CIFAR10, we focus on this strategy for STL10 and Tiny ImageNet. Table 1b and 1c shows the robust accuracy on STL10 and Tiny ImageNet using a ResNet18. Our approach also significantly outperforms regular AT on these datasets.

**Offline Test-time Adapattion.** As shown in Table 1a, 1b, 1c, the offline fine-tuning further improves the robust accuracy over the online version.

**Diverse Attacks.** Recommended by (Croce et al., 2022), in Appendix C.1, we evaluate our method on diverse attacks including *transfer attack*, *expectation attack* and *boundary attack*, where test-time adaptation all improves the robustness of the model.

## 5.2 METHOD ANALYSIS

We observe the significant positive correlation between the gradient of self-supervised loss $\mathcal{L}_{SS}$ and the classification loss $\mathcal{L}_{cls}$. Define

$$\rho(\widetilde{\mathbf{x}}_i^\star) = \frac{\nabla_{\boldsymbol{\theta}_E} \mathcal{L}_{cls}(\widetilde{\mathbf{x}}_i^\star, \widetilde{y}_i)^T \nabla_{\boldsymbol{\theta}_E} \mathcal{L}_{SS}(\widetilde{\mathbf{x}}_i^\star)}{\|\nabla_{\boldsymbol{\theta}_E} \mathcal{L}_{cls}(\widetilde{\mathbf{x}}_i^\star, \widetilde{y}_i)\|_2 \|\nabla_{\boldsymbol{\theta}_E} \mathcal{L}_{SS}(\widetilde{\mathbf{x}}_i^\star)\|_2} \ , \tag{13}$$

and approximate $\mathcal{L}_{cls}$ by the Taylor expansion

$$\mathcal{L}_{cls}(\widetilde{\mathbf{x}}_i^\star, \widetilde{y}_i; \boldsymbol{\theta}_E - \eta \nabla_{\boldsymbol{\theta}_E} \mathcal{L}_{SS}(\widetilde{\mathbf{x}}_i^\star)) - \mathcal{L}_{cls}(\widetilde{\mathbf{x}}_i^\star, \widetilde{y}_i; \boldsymbol{\theta}_E) \approx -\eta \rho(\widetilde{\mathbf{x}}_i^\star) \|\nabla_{\boldsymbol{\theta}_E} \mathcal{L}_{cls}(\widetilde{\mathbf{x}}_i^\star, \widetilde{y}_i)\|_2 \|\nabla_{\boldsymbol{\theta}_E} \mathcal{L}_{SS}(\widetilde{\mathbf{x}}_i^\star)\|_2 \ .$$

As $\boldsymbol{\theta}_E$ contains millions of parameters, its gradient norm is typically large. Therefore, gradient descent w.r.t. $\mathcal{L}_{SS}$ should act as a good substitute for optimizing $\mathcal{L}_{cls}$ when $\rho(\widetilde{\mathbf{x}}_i^\star)$ is significantly larger than 0. We further confirm this empirically. For all adversarial test inputs $\widetilde{\mathbf{x}}^\star \sim \widetilde{D}^\star$, we regard $\rho(\widetilde{\mathbf{x}}^\star)$ as a random variable and calculate its empirical statistics on the test set. Table 2 shows the empirical statistics of an adversarially-trained model on CIFAR10, and Figure 2 shows the *c.d.f.* of $\rho(\widetilde{\mathbf{x}}^\star)$. The mean of $\rho(\widetilde{\mathbf{x}}^\star)$ is indeed significantly larger than 0 and $P(\rho(\widetilde{\mathbf{x}}^\star) > 0)$ is larger than the robust accuracy of the adversarially-trained network (50%-60%), which implies that self-supervised test-time fine-tuning helps to correctly classify the adversarial test images.

We further provide the theoretical analysis in a linear model in Theorem B.1, which shows that the correlated gradient significantly strengthens the robustness and lowers natural risk. Besides, the correlated gradient also helps the model to move closer to the Bayesian robust estimator $\widehat{\boldsymbol{\theta}}_{\text{AB}}$.

## 5.3 ABLATION STUDY

**Meta Adversarial Training.** To show the effectiveness of MAT, we perform an ablation study to fine-tune the model with regular AT (*i.e.*, setting $\alpha = 0$ in line 5 of Algorithm 1). Table 7 shows that the robust accuracy and the improvements of fine-tuning are consistently worse without MAT.

**Accuracy Improvement on Inputs with Different Adversarial Budget.** As shown in Table 8, we set $\varepsilon = 0.015$ to perform the online test-time fine-tuning, showing that our method is also able to improve the robust accuracy of inputs with different adversarial budgets.

**Removing $\mathcal{L}_{SS}$ or $\mathcal{L}_R$.** To study the effect of $\mathcal{L}_{SS}$ and $\mathcal{L}_R$ in $\mathcal{L}_{test}$, we report the robust accuracy after online fine-tuning using only $\mathcal{L}_R$ and $\mathcal{L}_{SS}$ in Table 9. While, as expected, removing $\mathcal{L}_{SS}$ tends to reduce more accuracy than removing $\mathcal{L}_R$. It shows the benefits of our self-supervised test-time fine-tuning strategy. Nevertheless, the best results are obtained by exploiting both loss terms.

**Improvement on Clean Images.** As predicted by Theorem 3.1 and B.1, our method is able to improve not only the robust accuracy but also the natural accuracy. As shown in Table 10, our approach increases the clean image accuracy by test-time adaptation. This phenomenon further strengthens our conjecture that the improvement of robust accuracy is due to the improvement of generalization instead of gradient masking.

## 6 CONCLUSION

In linear models and two-layer random networks, we theoretically demonstrate the necessity of test-time adaptation for the model to achieve optimal robustness. To this end, we propose self-supervised test-time fine-tuning on adversarially-trained models to improve their generalization ability. Furthermore, we introduce a MAT strategy to find a good starting point for our self-supervised fine-tuning process. Our extensive experiments on CIFAR10, STL10 and Tiny ImageNet demonstrate that our method consistently improves the robust accuracy under different attack strategies, including strong adaptive attacks where the attacker is aware of our test-time adaptation technique. In these experiments, we utilize three different sources of self-supervision: rotation prediction, vertical flip prediction and the ensemble of them.

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

# A  PROOFS OF THEOREMS

## A.1  PRELIMINARY: MARCHENKO-PASTUR LAW AND TRANSFORMATION OF EIGENVALUES

Before the proof linear models, we first give the asymptotic spectrum of the matrix $\mathbf{K} = \mathbf{X}^\top \mathbf{X}$, and some useful results of the trace of the transformation of $\mathbf{K}$.

**Lemma A.1** (Marchenko-Pastur Law, Theorem 3.4 in (Bai & Silverstein, 2010)). *We define the eigenvalues of* $\mathbf{X}^\top \mathbf{X}$: $\lambda_1 > \lambda_2 > \cdots \lambda_n$ *has distribution with c.d.f.*

$$H_n(s) = \frac{1}{n} \sum_{i=1}^n \mathbf{1}_{\lambda_i \leq s}.$$

*If each dimension of* $\mathbf{x}$ *is i.i.d. with* $\mathbb{E}\mathbf{x} = 0$, $Cov(\mathbf{x}) = \mathbf{I}_d/d$ *and* $\mathbb{E}[\sqrt{d}x^i]^4 \leq M$ *for some universal constant* $M < \infty$, *then when* $n, d \to \infty$ *with* $n/d = c \in (0, 1)$, *for any bounded function* $g$, *when* $n, d \to \infty$ *with* $c = n/d \in [0, \infty)$

$$\int g(s)dH_n(s) \to \int g(s)dH(s),$$

*with p.d.f.* $dH(s)$

$$dH(s) = \frac{1}{2\pi c} \frac{\sqrt{(\lambda_+ - s)(s - \lambda_-)}}{s} \mathbf{1}_{s \in [\lambda_-, \lambda_+]} ds,$$

*where* $\lambda_- = (1 - \sqrt{c})^2$ *and* $\lambda_+ = (1 + \sqrt{c})^2$.

**Lemma A.2.** *If each dimension of* $\mathbf{x}$ *is i.i.d. with* $\mathbb{E}\mathbf{x} = 0$, $Cov(\mathbf{x}) = \mathbf{I}_d/d$ *and* $\mathbb{E}[\sqrt{d}x^i]^4 \leq M$ *for some universal constant* $M < \infty$, *then when* $n, d \to \infty$ *with* $n/d = c \in (0, 1)$,

$$\text{Tr}(\mathbf{K}(\mathbf{K} + \lambda \mathbf{I}_n)^{-2}) = d\frac{1 + c + \lambda - \sqrt{\Delta}}{2\sqrt{\Delta}},$$

$$\text{Tr}(\mathbf{K}^2(\mathbf{K} + \lambda \mathbf{I}_n)^{-2}) = d\frac{(1 + c + \lambda) - \sqrt{\Delta}}{2}(1 - \frac{\lambda}{\sqrt{\Delta}}),$$

$$\text{Tr}((\mathbf{K} + \lambda \mathbf{I}_n)^{-2}) = d\frac{(1 + c)(1 + c + \lambda) - 4c - (1 - c)\sqrt{\Delta}}{2\lambda^2\sqrt{\Delta}},$$

*where* $\Delta = (1 + c + z)^2 - 4c$.

*Proof.* We define three transformations of the eigenvalue of $dH(s)$

$$t_1(z) = \int \frac{s}{(s+z)^2} dH(s), \quad t_2(z) = \int \frac{s^2}{(s+z)^2} dH(s), \quad t_3(z) = \int \frac{1}{(s+z)^2} dH(s), \quad (14)$$

for $z \in [0, \infty)$. They can be calculated by the Stieltjes transformation of Marchenko-Pastur law.

By Marchenko-Pastur semicircular law, the Stieltjes transformation of $dH(s)$ is (Lemma 3.11 in (Bai & Silverstein, 2010) and Lemma 4.4 in (Cheng et al., 2022))

$$m(z) = \int \frac{1}{s - z} dH(s) = \frac{1 - c - z - \sqrt{(1 + c - z)^2 - 4c}}{2cz}$$

Let $\Delta = (1 + c + z)^2 - 4c$

$$t(z) = m(-z) = \frac{1 - c + z - \sqrt{(1 + c + z)^2 - 4c}}{-2cz},$$

and

$$\int \frac{1}{(s+z)^2} dH(s) = -\frac{d}{dz} \int \frac{1}{s+z} dH(s) = -\frac{dt(z)}{dz}.$$

Then

$$t_1(z) = \int \frac{1}{s+z} dH(s) - \int \frac{z}{(s+z)^2} dH(s) = t(z) + z\frac{dt(z)}{dz} = \frac{1+c+z-\sqrt{\Delta}}{2c\sqrt{\Delta}}, \qquad (15)$$

$$\begin{aligned} t_2(z) =& 1 - \int \frac{2z}{s+z} dH(s) + \int \frac{z^2}{(s+z)^2} dH(s) = 1 - 2zt(z) - z^2\frac{dt(z)}{dz} \\ =& \frac{(1+c+z)-\sqrt{\Delta}}{2c}(1 - \frac{z}{\sqrt{\Delta}}), \end{aligned} \qquad (16)$$

$$t_3(z) = -\frac{d}{dz} \int \frac{1}{s+z} dH(s) = \frac{dt(z)}{dz} = \frac{(1+c)(1+c+z) - 4c - (1-c)\sqrt{\Delta}}{2cz^2\sqrt{\Delta}}. \qquad (17)$$

The trace operation can be translated into:

$$\mathrm{Tr}(\mathbf{K}(\mathbf{K} + \lambda\mathbf{I}_n)^{-2}) = n \int \frac{s}{(s+\lambda)^2} dH_n(s),$$

$$\mathrm{Tr}(\mathbf{K}^2(\mathbf{K} + \lambda\mathbf{I}_n)^{-2}) = n \int \frac{s^2}{(s+\lambda)^2} dH_n(s),$$

$$\mathrm{Tr}((\mathbf{K} + \lambda\mathbf{I}_n)^{-2}) = n \int \frac{1}{(s+\lambda)^2} dH_n(s).$$

Therefore, when $n, d \to \infty$ with $c = n/d \in (0, 1)$,

$$\mathrm{Tr}(\mathbf{K}(\mathbf{K} + \lambda\mathbf{I}_n)^{-2}) \to n\frac{1+c+\lambda-\sqrt{\Delta}}{2c\sqrt{\Delta}} = d\frac{1+c+\lambda-\sqrt{\Delta}}{2\sqrt{\Delta}},$$

$$\mathrm{Tr}(\mathbf{K}^2(\mathbf{K} + \lambda\mathbf{I}_n)^{-2}) \to n\frac{(1+c+\lambda)-\sqrt{\Delta}}{2c}(1 - \frac{\lambda}{\sqrt{\Delta}}) = d\frac{(1+c+\lambda)-\sqrt{\Delta}}{2}(1 - \frac{\lambda}{\sqrt{\Delta}}),$$

$$\begin{aligned} \mathrm{Tr}((\mathbf{K} + \lambda\mathbf{I}_n)^{-2}) \to& n\frac{(1+c)(1+c+\lambda) - 4c - (1-c)\sqrt{\Delta}}{2c\lambda^2\sqrt{\Delta}} \\ =& d\frac{(1+c)(1+c+\lambda) - 4c - (1-c)\sqrt{\Delta}}{2\lambda^2\sqrt{\Delta}}. \end{aligned}$$

where $\Delta = (1+c+\lambda)^2 - 4c$. □

## A.2 PROOF OF THEOREM 3.1

We separate the proof into three parts for three estimators separately.

*Proof of $\widehat{\boldsymbol{\theta}}_{AT}^{Lin}$.* For $\widehat{\boldsymbol{\theta}}_{\mathrm{Lin}}^{\mathrm{AT}}$, taking gradient with respect to the objective function

$$\frac{1}{n} \sum_{i=1}^n \left[ (y_i - \mathbf{x}_i^\top \boldsymbol{\theta}) + \varepsilon^2 \|\boldsymbol{\theta}\|^2 \right],$$

we obtain

$$\frac{2}{n}(\mathbf{Y} - \mathbf{X}^\top \boldsymbol{\theta}) + 2n\varepsilon\boldsymbol{\theta} = 0.$$

Therefore,

$$\widehat{\boldsymbol{\theta}}_{\mathrm{AT}}^{\mathrm{Lin}} = \mathbf{X}(\mathbf{X}^\top \mathbf{X} + n\varepsilon^2 \mathbf{I}_n)^{-1}\mathbf{Y}$$

Its natural risk is

$$\begin{aligned} R_{\mathbf{x}}^{\mathrm{nat}}(\widehat{\boldsymbol{\theta}}_{\mathrm{AT}}^{\mathrm{Lin}}) =& \mathbb{E}_{\boldsymbol{\xi}, \boldsymbol{\theta}_*, \mathbf{x}}(\mathbf{x}^\top(\widehat{\boldsymbol{\theta}}_{\mathrm{AT}}^{\mathrm{Lin}} - \boldsymbol{\theta}_*))^2 = \frac{1}{d}\|\widehat{\boldsymbol{\theta}}_{\mathrm{AT}}^{\mathrm{Lin}} - \boldsymbol{\theta}_*\|^2 \\ =& \frac{1}{d}\mathbb{E}_{\boldsymbol{\xi}, \boldsymbol{\theta}_*}\|\boldsymbol{\theta}_* - \mathbf{X}(\mathbf{X}^\top \mathbf{X} + \lambda\mathbf{I}_n)^{-1}(\mathbf{X}^\top \boldsymbol{\theta}_* + \boldsymbol{\xi})\|^2 \\ =& \tau^2(1 - \frac{n}{d}) + \frac{\lambda^2\tau^2}{d}\mathrm{Tr}((\mathbf{K} + \lambda\mathbf{I}_n)^{-2}) + \frac{\sigma^2}{d}\mathrm{Tr}(\mathbf{K}(\mathbf{K} + \lambda\mathbf{I}_n)^{-2}) \end{aligned} \qquad (18)$$

And the Lipschitz constant is

$$
\begin{aligned}
L(\widehat{\boldsymbol{\theta}}_{\mathrm{AT}}^{\mathrm{Lin}})^2 =& \|\widehat{\boldsymbol{\theta}}_{\mathrm{AT}}^{\mathrm{Lin}}\|^2 \\
=& \mathbb{E}_{\boldsymbol{\xi},\boldsymbol{\theta}_*} \|(\mathbf{X}\mathbf{X}^\top + \lambda\mathbf{I}_d)^{-1}\mathbf{X}(\mathbf{X}^\top\boldsymbol{\theta}_* + \boldsymbol{\xi})\|^2 \\
=& \tau^2 \mathrm{Tr}\left((\mathbf{K} + \lambda\mathbf{I}_n)^{-2}\mathbf{K}^2\right) + \sigma^2 \mathrm{Tr}\left((\mathbf{K} + \lambda\mathbf{I}_n)^{-2}\mathbf{K}\right).
\end{aligned}
\tag{19}
$$

When $\lambda = n\varepsilon^2 \to \infty$, by Lemma A.2,

$$
\mathrm{Tr}(\mathbf{K}(\mathbf{K} + \lambda\mathbf{I}_n)^{-2}) \to 0, \quad \mathrm{Tr}(\mathbf{K}^2(\mathbf{K} + \lambda\mathbf{I}_n)^{-2}) \to 0, \quad \lambda^2 \mathrm{Tr}((\mathbf{K} + \lambda\mathbf{I}_n)^{-2}) \to \frac{n}{d}
\tag{20}
$$

Therefore,

$$
R_{\mathbf{x}}^{\mathrm{adv}}(\widehat{\boldsymbol{\theta}}_{\mathrm{AT}}^{\mathrm{Lin}}) = R_{\mathbf{x}}^{\mathrm{nat}}(\widehat{\boldsymbol{\theta}}_{\mathrm{AT}}^{\mathrm{Lin}}) + \varepsilon^2 L(\widehat{\boldsymbol{\theta}}_{\mathrm{AT}}^{\mathrm{Lin}}) \to \tau^2
\tag{21}
$$

$\square$

**Lemma A.3.** *If the oracle parameter $\boldsymbol{\theta}_*$ is independent of $\mathbf{x}$ and has the prior distribution $\boldsymbol{\theta}_* \sim \mathcal{N}(0, \tau^2\mathbf{I})$, and the noise $\xi \sim \mathcal{N}(0, \sigma^2)$. Furthermore, if we assume each dimension of $\mathbf{x}$ is i.i.d. with $\mathbb{E}\mathbf{x} = 0$, $Cov(\mathbf{x}) = \mathbf{I}_d/d$ and $\mathbb{E}[\sqrt{d}x^i]^4 \le M$ for some universal constant $M < \infty$, when $n, d \to \infty$ with $n/d = c \in (0, 1)$, then for $\lambda_* = \sigma^2/\tau^2$ and $\widehat{\boldsymbol{\theta}} = \frac{1}{A}\mathbf{X}(\mathbf{X}^\top\mathbf{X} + \lambda_*\mathbf{I}_n)Y$,*

$$
R_{\mathbf{x}}^{nat}(\widehat{\boldsymbol{\theta}}) = \tau^2 - \frac{\tau^2}{A}((1 + c + \lambda_*) - \sqrt{\Delta})(1 - \frac{1}{2A}),
$$

$$
L(\widehat{\boldsymbol{\theta}})^2 = \frac{d\tau^2}{2A^2}((1 + c + \lambda_*) - \sqrt{\Delta}),
$$

*where $\Delta = (1 + c + \lambda_*)^2 - 4c$.*

*Proof.*

$$
\begin{aligned}
R_{\mathbf{x}}^{\mathrm{nat}}(\widehat{\boldsymbol{\theta}}) =& \frac{1}{d}\mathbb{E}_{\boldsymbol{\xi},\boldsymbol{\theta}_*}\|\boldsymbol{\theta}_* - \frac{1}{A}\mathbf{X}(\mathbf{X}^\top\mathbf{X} + \lambda_*\mathbf{I}_n)^{-1}(\mathbf{X}^\top\boldsymbol{\theta}_* + \boldsymbol{\xi})\|^2 \\
=& \frac{\tau^2}{d}\mathrm{Tr}\left(\left(\mathbf{I}_d - \frac{1}{A}\mathbf{X}(\mathbf{K} + \lambda_*\mathbf{I}_n)^{-1}\mathbf{X}^\top\right)^2\right) + \frac{\sigma^2}{d}\mathrm{Tr}\left(\frac{1}{A^2}\mathbf{X}(\mathbf{K} + \lambda_*\mathbf{I}_n)^{-2}\mathbf{X}^\top\right) \\
=& \frac{\tau^2}{d}(d - \frac{2n}{A}) + \frac{2\tau^2\lambda_*}{dA}\mathrm{Tr}\left((\mathbf{K} + \lambda_*\mathbf{I}_n)^{-1}\right) + \frac{\tau^2}{dA^2}\mathrm{Tr}\left(\mathbf{K}^2(\mathbf{K} + \lambda_*\mathbf{I}_n)^{-2}\right) + \\
& \frac{\sigma^2}{dA^2}\mathrm{Tr}\left((\mathbf{K} + \lambda_*\mathbf{I}_n)^{-2}\mathbf{K}\right).
\end{aligned}
\tag{22}
$$

According to Lemma A.2, when $n, d \to \infty$ with $c = \frac{n}{d} \in (0, 1)$,

$$
\mathrm{Tr}((\mathbf{K} + \lambda_*\mathbf{I}_n)^{-1}) = d\frac{1 - c + \lambda_* - \sqrt{\Delta}}{-2\lambda_*}
$$

$$
\mathrm{Tr}(\mathbf{K}(\mathbf{K} + \lambda_*\mathbf{I}_n)^{-2}) = d\frac{1 + c + \lambda_* - \sqrt{\Delta}}{2\sqrt{\Delta}}
$$

$$
\mathrm{Tr}(\mathbf{K}^2(\mathbf{K} + \lambda_*\mathbf{I}_n)^{-2}) = d\frac{(1 + c + \lambda_*) - \sqrt{\Delta}}{2}(1 - \frac{\lambda_*}{\sqrt{\Delta}}),
$$

where $\lambda_* = \sigma^2/\tau^2$ and $\Delta = (1 + c + \lambda_*)^2 - 4c$. Therefore,

$$
R_{\mathbf{x}}^{\mathrm{nat}}(\widehat{\boldsymbol{\theta}}_{\mathrm{bayes}}^{\mathrm{lin}}) = \tau^2 - \frac{\tau^2}{A}((1 + c + \lambda_*) - \sqrt{\Delta})(1 - \frac{1}{2A}).
$$

And

$$
\begin{aligned}
L(\widehat{\boldsymbol{\theta}})^2 =& \mathbb{E}_{\boldsymbol{\xi},\boldsymbol{\theta}_*}\|\frac{1}{A}\mathbf{X}(\mathbf{X}^\top\mathbf{X} + \lambda_*\mathbf{I}_n)^{-1}(\mathbf{X}^\top\boldsymbol{\theta}_* + \boldsymbol{\xi})\|^2 \\
=& \frac{\tau^2}{A^2}\mathrm{Tr}\left((\mathbf{X}(\mathbf{K} + \lambda_*\mathbf{I}_n)^{-1}\mathbf{X}^\top)^2\right) + \frac{\sigma^2}{A^2}\mathrm{Tr}\left(\frac{1}{A^2}\mathbf{X}(\mathbf{K} + \lambda_*\mathbf{I}_n)^{-2}\mathbf{X}^\top\right) \\
=& \frac{\tau^2}{A^2}\mathrm{Tr}\left(\mathbf{K}^2(\mathbf{K} + \lambda_*\mathbf{I}_n)^{-2}\right) + \frac{\sigma^2}{A^2}\mathrm{Tr}\left((\mathbf{K} + \lambda_*\mathbf{I}_n)^{-2}\mathbf{K}\right).
\end{aligned}
\tag{23}
$$

Therefore,

$$L(\widehat{\boldsymbol{\theta}}) = \frac{d\tau^2}{2A^2}((1 + c + \lambda_*) - \sqrt{\Delta}). \tag{24}$$

$\square$

*Proof of $\widehat{\boldsymbol{\theta}}_{RB}^{Lin}$.* It is well known that the posterior distribution of $\boldsymbol{\theta}_*$ is

$$\boldsymbol{\theta}_*|\mathbf{X}, \mathbf{Y} \sim \mathcal{N}(\mathbf{X}(\mathbf{X}^\top\mathbf{X} + \lambda_*\mathbf{I}_n)^{-1}Y, \sigma^2(\mathbf{X}\mathbf{X}^\top + \lambda_*\mathbf{I}_m)^{-1}),$$

where $\lambda_* = \sigma^2/\tau^2$. According to the definition of Bayesian estimator,

$$\widehat{\boldsymbol{\theta}}_{\mathrm{RB}}^{\mathrm{Lin}} = \arg\min_{\widehat{\boldsymbol{\theta}}} \mathbb{E}_{\boldsymbol{\theta}_*|\mathbf{X}}\mathbb{E}_{\mathbf{x}}\left((\mathbf{x}^\top(\widehat{\boldsymbol{\theta}} - \boldsymbol{\theta}_*))^2 + \varepsilon^2\|\widehat{\boldsymbol{\theta}}\|^2\right).$$

As the linear model only allows fixed $\widehat{\boldsymbol{\theta}}$ for each $\mathbf{x}$, we obtain

$$\begin{aligned}\frac{d}{d\widehat{\boldsymbol{\theta}}}\mathbb{E}_{\boldsymbol{\theta}_*|\mathbf{X}}\left(\frac{1}{d}\|\widehat{\boldsymbol{\theta}} - \boldsymbol{\theta}_*\|^2 + \varepsilon^2\|\widehat{\boldsymbol{\theta}}\|^2\right) &= \mathbb{E}_{\boldsymbol{\theta}_*|\mathbf{X}}\frac{d}{d\widehat{\boldsymbol{\theta}}}\left(\frac{1}{d}\|\widehat{\boldsymbol{\theta}} - \boldsymbol{\theta}_*\|^2 + \varepsilon^2\|\widehat{\boldsymbol{\theta}}\|^2\right)\\ &= \mathbb{E}_{\boldsymbol{\theta}_*|\mathbf{X}}\left(\frac{2}{d}(\widehat{\boldsymbol{\theta}} - \boldsymbol{\theta}_*) + 2\varepsilon^2\widehat{\boldsymbol{\theta}}\right) = 0.\end{aligned} \tag{25}$$

Therefore,

$$\widehat{\boldsymbol{\theta}}_{\mathrm{RB}}^{\mathrm{Lin}} = \frac{1}{\varepsilon^2 d + 1}\mathbb{E}_{\boldsymbol{\theta}_*|\mathbf{X}}\boldsymbol{\theta}_* = \frac{1}{\varepsilon^2 d + 1}\mathbf{X}(\mathbf{X}^\top\mathbf{X} + \lambda_*\mathbf{I}_n)^{-1}Y.$$

Using Lemma A.3, when $d \to \infty$

$$\begin{aligned}R_{\mathbf{x}}^{\mathrm{nat}}(\widehat{\boldsymbol{\theta}}_{\mathrm{RB}}^{\mathrm{Lin}}) &= \tau^2 - \frac{\tau^2}{\varepsilon^2 d + 1}((1 + c + \lambda_*) - \sqrt{\Delta})\frac{2\varepsilon^2 d + 1}{2(\varepsilon^2 d + 1)} \to \tau^2,\\ L(\widehat{\boldsymbol{\theta}}_{\mathrm{RB}}^{\mathrm{Lin}})^2 &= \frac{d\tau^2}{2(\varepsilon^2 d + 1)^2}((1 + c + \lambda_*) - \sqrt{\Delta}) \to 0.\end{aligned} \tag{26}$$

Summarizing the results,

$$R_{\mathbf{x}}^{\mathrm{nat}}(\widehat{\boldsymbol{\theta}}_{\mathrm{RB}}^{\mathrm{Lin}}) = R_{\mathbf{x}}^{\mathrm{nat}}(\widehat{\boldsymbol{\theta}}_{\mathrm{RB}}^{\mathrm{Lin}}) + \varepsilon^2 L(\widehat{\boldsymbol{\theta}}_{\mathrm{RB}}^{\mathrm{Lin}})^2 \to \tau^2 \tag{27}$$

$\square$

*Proof of $\widehat{\boldsymbol{\theta}}_{AB}^{Lin}$.* Bayesian robust estimator of each $\mathbf{x}$, which optimizes $(\mathbf{x}^\top(\widehat{\boldsymbol{\theta}} - \boldsymbol{\theta}_*))^2 + \varepsilon^2\|\widehat{\boldsymbol{\theta}}\|^2$ is

$$\widehat{\boldsymbol{\theta}}_{\mathrm{AB}}^{\mathrm{Lin}} = \arg\min_{\widehat{\boldsymbol{\theta}}} \mathbb{E}_{\boldsymbol{\theta}_*|\mathbf{X}}\left((\mathbf{x}^\top(\widehat{\boldsymbol{\theta}} - \boldsymbol{\theta}_*))^2 + \varepsilon^2\|\widehat{\boldsymbol{\theta}}\|^2\right),$$

where it is well known that the posterior distribution of $\boldsymbol{\theta}_*$ is

$$\boldsymbol{\theta}_*|\mathbf{X}, \mathbf{Y} \sim \mathcal{N}(\mathbf{X}(\mathbf{X}^\top\mathbf{X} + \lambda_*\mathbf{I}_n)^{-1}Y, \sigma^2(\mathbf{X}\mathbf{X}^\top + \lambda_*\mathbf{I}_m)^{-1}).$$

Taking the gradient w.r.t $\widehat{\boldsymbol{\theta}}$ gives the solution

$$\begin{aligned}\widehat{\boldsymbol{\theta}}_{\mathrm{AB}}^{\mathrm{Lin}}(\mathbf{x}) &= (\mathbf{x}\mathbf{x}^\top + \varepsilon^2\mathbf{I}_d)^{-1}\mathbf{x}\mathbf{x}^\top\mathbf{X}(\mathbf{X}^\top\mathbf{X} + \lambda_*\mathbf{I}_n)^{-1}Y\\ &= \frac{\mathbf{x}\mathbf{x}^\top}{\varepsilon^2 + \mathbf{x}^\top\mathbf{x}}\widehat{\boldsymbol{\theta}}_{\mathrm{nat}}^{\mathrm{Lin}},\end{aligned} \tag{28}$$

where $\widehat{\boldsymbol{\theta}}_{\mathrm{nat}}^{\mathrm{Lin}} = \mathbf{X}(\mathbf{X}^\top\mathbf{X} + \lambda_*\mathbf{I}_n)^{-1}(\mathbf{X}^\top\boldsymbol{\theta}_* + \boldsymbol{\xi})$ with $\lambda_* = \sigma^2/\tau^2$ is the Bayesian estimator for natural risk.

For its natural risk $R_{\mathbf{x}}^{\mathrm{nat}}(\widehat{\boldsymbol{\theta}}_{\mathrm{AB}}^{\mathrm{Lin}})$,

$$R_{\mathbf{x}}^{\mathrm{nat}}(\widehat{\boldsymbol{\theta}}_{\mathrm{AB}}^{\mathrm{Lin}}) = \mathbb{E}_{\boldsymbol{\theta}_*,\boldsymbol{\xi}}\mathbb{E}_{\mathbf{x}}(\mathbf{x}^\top(\widehat{\boldsymbol{\theta}}_{\mathrm{AB}}^{\mathrm{Lin}} - \boldsymbol{\theta}_*))^2$$

As

$$\mathbf{x}^\top\widehat{\boldsymbol{\theta}} = \frac{\mathbf{x}^\top\mathbf{x}\mathbf{x}^\top}{\varepsilon^2 + \mathbf{x}^\top\mathbf{x}}\widehat{\boldsymbol{\theta}}_{\mathrm{nat}}^{\mathrm{Lin}} = \mathbf{x}^\top\left(\frac{\mathbf{x}^\top\mathbf{x}}{\varepsilon^2 + \mathbf{x}^\top\mathbf{x}}\widehat{\boldsymbol{\theta}}_{\mathrm{nat}}^{\mathrm{Lin}}\right),$$

then

$$R_{\mathbf{x}}^{\text{nat}}(\widehat{\boldsymbol{\theta}}_{\text{AB}}^{\text{Lin}}) = \mathbb{E}_{\boldsymbol{\theta}_*, \boldsymbol{\xi}} \mathbb{E}_{\mathbf{x}}(\mathbf{x}^\top (\frac{\mathbf{x}^\top \mathbf{x}}{\varepsilon^2 + \mathbf{x}^\top \mathbf{x}} \widehat{\boldsymbol{\theta}}_{\text{nat}}^{\text{Lin}} - \boldsymbol{\theta}_*))^2.$$

When $d \to \infty$, $\mathbf{x}^\top \mathbf{x} \to 1$ in probability. Therefore,

$$\frac{\mathbf{x}^\top \mathbf{x}}{\varepsilon^2 + \mathbf{x}^\top \mathbf{x}} \to \frac{1}{1 + \varepsilon^2}. \tag{29}$$

Then when $d \to \infty$,

$$R_{\mathbf{x}}^{\text{nat}}(\widehat{\boldsymbol{\theta}}_{\text{AB}}^{\text{Lin}}) = \mathbb{E}_{\boldsymbol{\theta}_*, \boldsymbol{\xi}} \| \frac{1}{\varepsilon^2 + 1} \widehat{\boldsymbol{\theta}}_{\text{nat}}^{\text{Lin}} - \boldsymbol{\theta}_* \|^2$$

By Lemma A.3,

$$R_{\mathbf{x}}^{\text{nat}}(\widehat{\boldsymbol{\theta}}_{\text{AB}}^{\text{Lin}}) = \tau^2 - \frac{\tau^2}{\varepsilon^2 + 1}((1 + c + \lambda_*) - \sqrt{\Delta})(1 - \frac{1}{2(\varepsilon^2 + 1)}) \tag{30}$$

As $\mathbf{x}^\top \mathbf{x} \to 1$ in probability when $d \to \infty$, $\frac{\mathbf{x}^\top \mathbf{x}}{(\varepsilon^2 + \mathbf{x}^\top \mathbf{x})^2} \to \frac{1}{(\varepsilon^2 + 1)^2}$. Then

$$L(\widehat{\boldsymbol{\theta}}_{\text{AB}}^{\text{Lin}})^2 \to \frac{1}{d(\varepsilon^2 + 1)^2} \mathbb{E}_{\boldsymbol{\theta}_*, \boldsymbol{\xi}} \| \widehat{\boldsymbol{\theta}}_{\text{nat}}^{\text{Lin}} \|^2. \tag{31}$$

From Lemma A.3,

$$\mathbb{E}_{\boldsymbol{\theta}_*, \boldsymbol{\xi}} \| \widehat{\boldsymbol{\theta}}_{\text{nat}}^{\text{Lin}} \|^2 = d\tau^2 \frac{(1 + c + \lambda_*) - \sqrt{\Delta}}{2}.$$

Summarizing two parts, the adversarial risk is

$$R_{\mathbf{x}}^{\text{adv}}(\widehat{\boldsymbol{\theta}}_{\text{AB}}^{\text{Lin}}) = \tau^2 - \frac{\tau^2}{2(\varepsilon^2 + 1)}((1 + c + \lambda_*) - \sqrt{\Delta}) \leq \tau^2(1 - \frac{c}{(\varepsilon^2 + 1)(1 + c + \lambda_*)}).$$

$\square$

### A.3 PROOF OF THEOREM 3.2

*Proof.* As $\widehat{\boldsymbol{\theta}}_{\text{AB}}^{\text{Lin}} = \frac{\mathbf{x}\mathbf{x}^\top \widehat{\boldsymbol{\theta}}_{\text{nat}}^{\text{Lin}}}{\varepsilon^2 + \mathbf{x}^\top \mathbf{x}}$, the adversarial input

$$\mathbf{x}^\star = \varepsilon \frac{\widehat{\boldsymbol{\theta}}_{\text{AB}}^{\text{Lin}}}{\| \widehat{\boldsymbol{\theta}}_{\text{AB}}^{\text{Lin}} \|} = \mathbf{x} + \varepsilon \frac{\mathbf{x}}{\| \mathbf{x} \|}$$

As $d \to \infty$, $\| \mathbf{x} \| \to 1$ in probability. Therefore, $\mathbf{x}^\star = (1 + \varepsilon)\mathbf{x}$. Taking it into

$$\widehat{\boldsymbol{\theta}}_{\text{AB},\star}^{\text{Lin}} = (\mathbf{x}^\star \mathbf{x}^{\star\top} + \varepsilon^2 \mathbf{I}_d)^{-1} \mathbf{x}^\star \mathbf{x}^{\star\top} \mathbf{X} (\mathbf{X}^\top \mathbf{X} + \lambda_* \mathbf{I}_n)^{-1} \mathbf{Y},$$

we obtain

$$\widehat{\boldsymbol{\theta}}_{\text{AB},\star}^{\text{Lin}} = \frac{(1 + \varepsilon)^2 \mathbf{x}\mathbf{x}^\top \widehat{\boldsymbol{\theta}}_{\text{nat}}^{\text{Lin}}}{\varepsilon + (1 + \varepsilon)^2 \mathbf{x}^\top \mathbf{x}}$$

From Eqn (30) and 31,

$$R_{\mathbf{x}}^{\text{nat}}(\widehat{\boldsymbol{\theta}}_{\text{AB},\star}^{\text{Lin}}) = \tau^2 \left( 1 - \frac{(1 + \varepsilon)^2}{\varepsilon^2 + (1 + \varepsilon)^2}(1 + c + \lambda_* - \sqrt{\Delta}) \right) +$$

$$\frac{\tau^2 (1 + \varepsilon)^4}{2(\varepsilon^2 + (1 + \varepsilon)^2)^2}(1 + c + \lambda_* - \sqrt{\Delta})$$

$$L(\widehat{\boldsymbol{\theta}}_{\text{AB},\star}^{\text{Lin}})^2 = \frac{\tau^2 (1 + \varepsilon)^4}{2(\varepsilon^2 + (1 + \varepsilon)^2)^2}(1 + c + \lambda_* - \sqrt{\Delta}).$$

Therefore,

$$R_{\mathbf{x}}^{\text{adv}}(\widehat{\boldsymbol{\theta}}_{\text{AB},\star}^{\text{Lin}}) = R_{\mathbf{x}}^{\text{nat}}(\widehat{\boldsymbol{\theta}}_{\text{AB},\star}^{\text{Lin}}) + \varepsilon^2 L(\widehat{\boldsymbol{\theta}}_{\text{AB},\star}^{\text{Lin}})^2 = \tau^2 \left( 1 - (1 - \varepsilon^2 + \frac{2\varepsilon^2}{(1 + \varepsilon)^2}) \frac{1 + c + \lambda_* - \sqrt{\Delta}}{2(\varepsilon^2/(1 + \varepsilon)^2 + 1)} \right)$$

$\square$

## B  CORRELATED GRADIENTS

In the following Theorem, we show that with correlated gradient, one gradient descent step like our method largely improves the natural and adversarial risk of the model.

**Theorem B.1.** *We assume the oracle parameter $\boldsymbol{\theta}_*$ is independent of $\mathbf{x}$ and has the prior distribution $\boldsymbol{\theta}_* \sim \mathcal{N}(0, \tau^2 \mathbf{I})$, and the noise $\xi \sim \mathcal{N}(0, \sigma^2)$. Let $\widehat{\boldsymbol{\theta}}^0 = \widehat{\boldsymbol{\theta}}_{AT}$ be the estimator of adversarial training of the linear model $F^{Lin}(\mathbf{x}; \boldsymbol{\theta})$:*

$$\widehat{\boldsymbol{\theta}}^0 = \mathbf{X}(\mathbf{X}^\top \mathbf{X} + n\varepsilon^2 \mathbf{I}_n)^{-1} \mathbf{Y}.$$

*When receiving a new test data point $(\mathbf{x}^\star, y)$ and taking one gradient descent step with correlated gradient $\widehat{g}$: $\widehat{\boldsymbol{\theta}}^1 = \widehat{\boldsymbol{\theta}}^0 - \eta \widehat{\mathbf{g}}$, where $\|\mathbf{x}^\star - \mathbf{x}\| \leq \varepsilon$ is an adversarial example of $\mathbf{x}$, $Corr(\widehat{\mathbf{g}}, \nabla_{\widehat{\boldsymbol{\theta}}^0} \mathcal{L}(F^{Lin}(\mathbf{x}, \widehat{\boldsymbol{\theta}}^0), y)) = \rho > 0$ and $\eta$ is the learning rate. Let $\mathbf{x} = (x^1, \cdots, x^d)$ where $x^i$ is the $i$-th element of $\mathbf{x}$. We further assume that $\{x^i\}_{i=1}^d$ are i.i.d. with $\mathbb{E}[x^i] = 0$, $Var[x^i] = 1/d$. And $\mathbb{E}[\sqrt{d}x^i]^4 \leq M$ for some universal constant $M < \infty$. When $n, d \to \infty$ with $c = n/d \in (0, 1)$, with the optimal learning rate,*

$$R_{\mathbf{x}}^{nat}(\widehat{\boldsymbol{\theta}}^0) - R_{\mathbf{x}}^{nat}(\widehat{\boldsymbol{\theta}}^1) \geq \frac{\tau^2 \rho^2}{(((1+\varepsilon)^2 + \sigma^2/\tau^2)(1+\varepsilon)^2(1+\varepsilon)^2)^2} -$$

$$\frac{2\tau^2 \rho^2}{((1+\varepsilon)^2 + \sigma^2/\tau^2)(1+\varepsilon)^2(1+\varepsilon)^2}$$

$$R_{\mathbf{x}}^{adv}(\widehat{\boldsymbol{\theta}}^0) - R_{\mathbf{x}}^{adv}(\widehat{\boldsymbol{\theta}}^1) \geq \frac{\tau^2 \rho^2 (1-\varepsilon)^2}{((1+\varepsilon)^2 + \sigma^2/\tau^2)(1+\varepsilon)^2(1+\varepsilon^2)}.$$

**Remarks.** With correlated gradients, improvements of $R_{\mathbf{x}}^{nat}$ and $R_{\mathbf{x}}^{adv}$ are both positive when having $\rho > 0$. By taking correlated gradient descent on the parameter, we get large improvements of both natural and adversarial risks.

Theorem B.1 shows that fine-tuning with correlated gradient largely improves both clean performance and robustness of the models. In addition, for linear models, the Bayesian optimal estimator is $\widehat{\boldsymbol{\theta}}_{AB}^{Lin} \parallel \mathbf{x}$. And $\widehat{\boldsymbol{\theta}}^1 \parallel \widehat{\mathbf{g}}$ with $Corr(\mathbf{x}^\star, \widehat{\mathbf{g}}) = \rho$. As $\mathbf{x}^\star$ is close to $\mathbf{x}$, with a proper learning rate, we can get close to Bayesian robust estimator with correlated gradient descent.

### B.1  PROOF OF THEOREM B.1

*Proof.* For a new test input $(\mathbf{x}^\star, y)$, where $\|\mathbf{x}^\star - \mathbf{x}\| \leq \varepsilon$ is an adversarial example near the input, its MSE loss is

$$\mathcal{L}(\mathbf{x}, y, \boldsymbol{\theta}) = \frac{1}{2}(\mathbf{x}^{\star\top} \boldsymbol{\theta}_* + \xi - \mathbf{x}^{\star\top} \boldsymbol{\theta})^2.$$

Taking the gradient w.r.t $\boldsymbol{\theta}$

$$\nabla \mathcal{L}_{\boldsymbol{\theta}}(\mathbf{x}^\star, y, \boldsymbol{\theta}) = (\mathbf{x}^{\star\top} \boldsymbol{\theta} - \mathbf{x}^{\star\top} \boldsymbol{\theta}_* - \xi)\mathbf{x}^\star.$$

Suppose the self-supervised task gives a correlated version of gradient and updates $\widehat{\boldsymbol{\theta}}^0$ with one step of gradient descent

$$\widehat{\boldsymbol{\theta}}^1 = \widehat{\boldsymbol{\theta}}^0 - \eta[(\mathbf{x}^{\star\top} \widehat{\boldsymbol{\theta}}^0 - \mathbf{x}^{\star\top} \boldsymbol{\theta}_* - \xi)\widehat{\mathbf{g}}],$$

where $Corr(\widehat{\mathbf{g}}, \mathbf{x}^\star) = \rho$ and $\mathbb{E}[\widehat{\mathbf{g}}^\top \widehat{\mathbf{g}}] = \mathbb{E}[\mathbf{x}^{\star\top} \mathbf{x}^\star]$. From Theorem 3.1, when $\lambda = n\varepsilon^2 \to \infty$, $R_{\mathbf{x}}^{nat}(\widehat{\boldsymbol{\theta}}^0)$ and $\|\widehat{\boldsymbol{\theta}}^0\|^2$ can be simplified as:

$$\mathbb{E}_{\boldsymbol{\theta}_*, \xi} \|\widehat{\boldsymbol{\theta}}^0\|^2 = \frac{\tau^2(1+c) + \sigma^2}{\varepsilon^4 d} + o(\frac{1}{d}) \to 0, \quad R_{\mathbf{x}}^{nat}(\widehat{\boldsymbol{\theta}}^0) = \mathbb{E}_{\xi, \boldsymbol{\theta}_*} \|\widehat{\boldsymbol{\theta}}^0 - \boldsymbol{\theta}_*\|^2 \to \tau^2.$$

Therefore, when $d \to \infty$

$$\widehat{\boldsymbol{\theta}}^1 \to \eta[(\mathbf{x}^{\star\top} \boldsymbol{\theta}_* + \xi)\widehat{\mathbf{g}}].$$

Then when $d \to \infty$,

$$\mathbb{E}_{\mathbf{x}} \|\widehat{\boldsymbol{\theta}}^1\|^2 = \eta^2 \boldsymbol{\theta}_*^\top \mathbb{E}_{\mathbf{x}}[\widehat{\mathbf{g}}^\top \widehat{\mathbf{g}} \mathbf{x}^{\star\top} \mathbf{x}^\star] \boldsymbol{\theta}_* + \eta^2 \xi^2 \mathbb{E}_{\mathbf{x}}[\widehat{\mathbf{g}}^\top \widehat{\mathbf{g}}] + 2\eta^2 \xi \boldsymbol{\theta}_*^\top \mathbb{E}_{\mathbf{x}}[\widehat{\mathbf{g}}^\top \mathbf{x}^\star].$$

Therefore,
$$\mathbb{E}_{\mathbf{x},\boldsymbol{\theta}_*,\boldsymbol{\xi}}\|\widehat{\boldsymbol{\theta}}^1\|^2 = \eta^2 \left(\tau^2 \mathbb{E}_{\mathbf{x}}[\hat{\mathbf{g}}^\top \hat{\mathbf{g}} \mathbf{x}^{\star\top} \mathbf{x}^\star] + \sigma^2 \mathbb{E}_{\mathbf{x}}[\hat{\mathbf{g}}^\top \hat{\mathbf{g}}]\right).$$

By decomposing $\hat{\mathbf{g}} = \rho \mathbf{x}^\star + \sqrt{1-\rho^2}\mathbf{z}_*$, we can obtain,
$$\mathbb{E}_{\mathbf{x},\boldsymbol{\theta}_*,\boldsymbol{\xi}}\|\widehat{\boldsymbol{\theta}}^1\|^2 = \eta^2 \left(\tau^2 \mathbb{E}_{\mathbf{x}}[(\mathbf{x}^{\star\top}\mathbf{x}^\star)^2] + \sigma^2 \mathbb{E}_{\mathbf{x}}[\mathbf{x}^{\star\top}\mathbf{x}^\star]\right).$$

As $\|\mathbf{x}^\star - \mathbf{x}\| \le \varepsilon$,
$$\mathbb{E}_{\mathbf{x},\boldsymbol{\theta}_*,\boldsymbol{\xi}}\|\widehat{\boldsymbol{\theta}}^1\|^2 \le \eta^2 \left(\tau^2(1+\varepsilon)^4 + \sigma^2(1+\varepsilon)^2\right).$$

For natural risk
$$\begin{aligned}
R_{\mathbf{x}}^{\mathrm{nat}}(\widehat{\boldsymbol{\theta}}^1) =& \mathbb{E}_{\mathbf{x},\boldsymbol{\xi},\boldsymbol{\theta}_*}\|\mathbf{x}^\top(\boldsymbol{\theta}_* - \widehat{\boldsymbol{\theta}}^1)\|^2 \\
& \to \tau^2 + \eta^2\tau^2 \mathbb{E}_{\mathbf{x}}[\mathbf{x}^{\star\top}\mathbf{x}^\star(\mathbf{x}^\top \hat{g})^2] + \eta^2\sigma^2 \mathbb{E}_{\mathbf{x}}[(\mathbf{x}^\top \hat{g})^2] - 2\eta\tau^2\mathbb{E}[\mathbf{x}^\top\mathbf{x}^\star\mathbf{x}^\top\hat{\mathbf{g}}].
\end{aligned}$$

By $\|\mathbf{x}^\star - \mathbf{x}\| \le \varepsilon$, $\mathrm{Corr}(\hat{\mathbf{g}},\mathbf{x}^\star) = \rho$ and $\mathbb{E}[\hat{\mathbf{g}}^\top\hat{\mathbf{g}}] = \mathbb{E}[\mathbf{x}^{\star\top}\mathbf{x}^\star]$,
$$R_{\mathbf{x}}^{\mathrm{nat}}(\widehat{\boldsymbol{\theta}}^1) \le \tau^2 + \eta^2\tau^2(1+\varepsilon)^4 + \eta^2\sigma^2(1+\varepsilon)^2 - 2\eta\tau^2\rho(1-\varepsilon)^2.$$

Therefore,
$$R_{\mathbf{x}}^{\mathrm{adv}}(\widehat{\boldsymbol{\theta}}^1) \le \tau^2 + \eta^2\tau^2(1+\varepsilon^2)(1+\varepsilon)^4 + \eta^2\sigma^2(1+\varepsilon^2)(1+\varepsilon)^2 - 2\eta\tau^2\rho(1-\varepsilon)^2.$$

Optimizing $\eta$ get
$$\eta_* = \frac{\rho\tau^2(1-\varepsilon)^2}{(\tau^2(1+\varepsilon)^4 + \sigma^2(1+\varepsilon)^2)(1+\varepsilon^2)}. \tag{32}$$

With $\eta_*$,
$$R_{\mathbf{x}}^{\mathrm{adv}}(\widehat{\boldsymbol{\theta}}^1) \le \tau^2\left(1 - \frac{\rho^2(1-\varepsilon)^2}{((1+\varepsilon)^2 + \sigma^2/\tau^2)(1+\varepsilon)^2(1+\varepsilon^2)}\right), \tag{33}$$

and
$$R_{\mathbf{x}}^{\mathrm{nat}}(\widehat{\boldsymbol{\theta}}^1) \le \tau^2 - \frac{\tau^2\rho^2}{(((1+\varepsilon)^2 + \sigma^2/\tau^2)(1+\varepsilon)^2(1+\varepsilon^2))^2} - \frac{2\tau^2\rho^2}{((1+\varepsilon)^2 + \sigma^2/\tau^2)(1+\varepsilon)^2(1+\varepsilon^2)^2}. \tag{34}$$

Compared with $\widehat{\boldsymbol{\theta}}^0$
$$R_{\mathbf{x}}^{\mathrm{adv}}(\widehat{\boldsymbol{\theta}}^0) = \tau^2, \quad R_{\mathbf{x}}^{\mathrm{nat}}(\widehat{\boldsymbol{\theta}}^0) = \tau^2,$$

we have improvements of
$$\begin{aligned}
& R_{\mathbf{x}}^{\mathrm{nat}}(\widehat{\boldsymbol{\theta}}^0) - R_{\mathbf{x}}^{\mathrm{nat}}(\widehat{\boldsymbol{\theta}}^1) \\
& \ge \frac{\tau^2\rho^2}{(((1+\varepsilon)^2 + \sigma^2/\tau^2)(1+\varepsilon)^2(1+\varepsilon^2))^2} - \frac{2\tau^2\rho^2}{((1+\varepsilon)^2 + \sigma^2/\tau^2)(1+\varepsilon)^2(1+\varepsilon^2)}
\end{aligned} \tag{35}$$

and
$$R_{\mathbf{x}}^{\mathrm{adv}}(\widehat{\boldsymbol{\theta}}^0) - R_{\mathbf{x}}^{\mathrm{adv}}(\widehat{\boldsymbol{\theta}}^1) \ge \frac{\tau^2\rho^2(1-\varepsilon)^2}{((1+\varepsilon)^2 + \sigma^2/\tau^2)(1+\varepsilon)^2(1+\varepsilon^2)}. \tag{36}$$

$\square$

## C   ADDITIONAL EXPERIMENTS

### C.1   DIVERSE ATTACKS

**Transfer Attack.** In Table 3, we perform a transfer attack from the static adversarial defense. We use the robust networks with the same architecture as the substitute model, and the test-time adaptation also improves the robust accuracy.

**Expectation Attack.** In Table 4, we show the results of the expectation attack. We modify the adaptive attack and average the gradient from 10 fine-tuned models, whose training batches are different. We evaluate the model using the ensemble of rotation and vertical flip as the self-supervised task on CIFAR10. We evaluate the model with Adaptive-AutoPGD-EOT and Adaptive-SquareAttack-EOT. One is the strongest attack in our method and the other is a black-box attack that is less likely to be

Table 3: Accuracy on transfer attack on CIFAR10.

| Methods | Rotation | VFlip | Rotation + VFlip |
|---|---|---|---|
| Without FT | 84.77% | 86.55% | 86.36% |
| Online FT | **86.10%** | **87.00%** | **87.10%** |

Table 4: Accuracy on expectation attack on CIFAR10 with the ensemble of rotation and vertical flip task.

| Attacks | w/o Fine-tuning | w/ Fine-tuning |
|---|---|---|
| Adaptive-AutoPGD | 53.99% | 57.70% |
| Adaptive-AutoPGD-EOT | 53.99% | 57.65% |
| Adaptive-SquareAttack | 65.75% | 66.80% |
| Adaptive-SquareAttack-EOT | 65.75% | 66.73% |

affected by gradient masking. The experiment shows that the expectation attack has little influence on the improvement of our test-time adaptation.

**Boundary Attack.** We use one of the SOTA decision-based attacks: RayS (Chen & Gu, 2020). We test it on CIFAR10 with the ensemble of rotation and vertical flip. Table 5 shows that our method also improves the robust accuracy of the decision-based attack.

GMSA (Chen et al., 2021a) with AutoPGD. GMSA is a recently proposed attack algorithm targeted at the test-time model adaptation. We use the GMSA with AutoPGD to attack our method, and the results are shown in Table 6. Under GMSA, our test-time adaptation still significantly improves the robust accuracy. Moreover, Table 6 also demonstrates the strength of our adaptive attack strategy as it achieves a higher success rate than GMSA.

### C.2 ABLATION STUDY

**Meta Adversarial Training.** Our meta training strategy in Algorithm 1 aims to strengthen the correlation between the self-supervised tasks and classification. To show its effectiveness, we perform an ablation study where we fine-tune the model with regular AT (*i.e.*, setting $\alpha = 0$ in line 5 of Algorithm 1). We then perform the same test-time fine-tuning on the model without MAT, using the same hyperparameters as in the MAT case. As shown in Table 7, the robust accuracy and the improvements of fine-tuning are consistently worse without MAT.

**Accuracy Improvement on Inputs with Different Adversarial Budget.** Our method is also able to improve the robust accuracy of inputs with different adversarial budgets. As shown in Table 8, we set $\ell_\infty$ budget of the adversarial inputs to be $0.015$ to perform the online test-time fine-tuning. The robust accuracy is ed improved.

**Removing $\mathcal{L}_{SS}$ or $\mathcal{L}_R$.** In our previous experiments, test-time fine-tuning was achieved using a combination of two loss functions: $\mathcal{L}_{SS}$ and $\mathcal{L}_R$. To study the effect of each of these terms separately, we remove either one of them from $\mathcal{L}_{test}$. In Table 9, we report the robust accuracy after online fine-tuning using only $\mathcal{L}_R$ and only $\mathcal{L}_{SS}$. While, as expected, removing $\mathcal{L}_{SS}$ tends to reduce more accuracy than removing $\mathcal{L}_R$. It shows the benefits of our self-supervised test-time fine-tuning strategy. Nevertheless, the best results are obtained by exploiting both loss terms.

**Accuracy Improvement on Clean Images.** As shown in Eqn (30) and Theorem B.1, our method is able to improve not only the robust accuracy but also the natural accuracy of clean images on adversarially-trained models. To evidence this, we maintain all the components of our model and simply replace the adversarial input images with clean images (*i.e.* replacing $\widetilde{B}^\star$ with clean inputs $\widetilde{B}$ in Algorithm 2) and perform the same self-supervised test-time fine-tuning.

As shown in Table 10, our approach increases the clean image accuracy. This phenomenon further strengthens our conjecture that the improvement of robust accuracy is due to the improvement of generalization instead of perturbing the model parameters, because randomly perturbing the parameters usually lowers the natural accuracy of the model.

Table 5: Accuracy on RayS on CIFAR10 with the ensemble of rotation and vertical flip task.

| Attacks | w/o Fine-tuning | w/ Fine-tuning |
|---|---|---|
| RayS | 65.61% | 77.38% |
| Adaptive-RayS | - | 75.03% |

Table 6: Accuracy under GMSA on CIFAR10.

| Tasks | Rotation | VFlip | Rotation+VFlip |
|---|---|---|---|
| Meta AT w/o FT | 53.90% | 52.79% | 53.16% |
| Meta AT w/ FT + Adaptive AutoPGD | 56.96% | 56.79% | 57.70% |
| Meta AT w/ FT + GMSA AutoPGD | 57.60% | 58.53% | 59.63% |

**Attacking Objectives.** The improvement of the test-time adaptation is not affected by the attack objectives. Even if no information of the ground truth label is incorporated in the attack, the test-time adaptation improves the robust accuracy. When the attacker randomly lowers the score of the false label to perform the adversarial attack, if our method uses the information of the leaked label to improve the robust accuracy, it will predict the false label and reduce the accuracy. However, as shown in Table 11, the self-supervised test-time fine-tuning improves the robust accuracy on these "adversarial" images. Besides, previous experiments on clean images already show that test-time fine-tuning is effective even if there is no information of the ground truth label.

### C.3 ADDITIONAL COMPARISON

**Comparison with SOAP (Shi et al., 2020).** Our method is different from SOAP as we are fine-tuning the model to adapt to new examples instead of purifying the input. We apply SOAP-RP to the adversarially-trained model and find that its improvement is marginal. Under AutoPGD, the accuracy is improved from 53.09% to 53.57%. This improvement is much smaller than our method, whose improvement is from 53.09% to 57.93%. SOAP only has little effect when combined with the commonly used AT.

**Combination with (Gowal et al., 2020).** We combine our test-time adaptation with AT using additional data (Gowal et al., 2020). We apply our Meta AT to it with the ensemble of rotation and vertical flip. Using a WideResNet-28-10, it achieves a robust accuracy of 62.07% under AutoPGD. With our test-time adaptation, the robust accuracy is improved to 64.34%. The improvement of robust accuracy is 2.27%.

**Robust Accuracy v.s Fine-tuning Steps.** Figure 3 shows the robust accuracy at each step of the test-time fine-tuning for different self-supervised tasks and attack methods. When using the standard version of attacks, the robust accuracy gradually increases as fine-tuning proceeds. When using our adaptive attacks, the adversarial examples are generated to attack the network with $\theta^T$ ($T = 10$) instead of $\theta^0$. Thus, when the parameters gradually change from $\theta^0$ to $\theta^T$, the accuracy drops.

**Inference Time.** Table 12 shows the inference time for different methods. While the inference time for our method is larger than SOAP and the normal method when the batch size is 1, the inference time gets closer when using a larger batch size. And the batch size of 20 or more is a common scenario of the inference. In order to achieve the statistical optimal adversarial risk, additional time

Table 7: Ablation study on the online test-time fine-tuning. The dataset is CIFAR10 and the task is the "Rotation + VFlip". All attacks are standard attacks. SA stands for Square Attack.

| Methods | SA | PGD-20 | AutoPGD | FAB |
|---|---|---|---|---|
| Regular AT | 65.64% | 59.19% | 53.16% | 53.05% |
| Online FT | 66.26% | 60.18% | 56.86% | 75.26% |
| Improvement | 0.62% | 0.99% | 3.70% | 22.21% |
| Meta AT | 65.75% | 59.51% | 53.99% | 53.85% |
| Online FT | 67.34% | 61.79% | 59.23% | 76.39% |
| Improvement | **1.59%** | **2.28%** | **5.24%** | **22.54%** |

Table 8: Robust test accuracy on CIFAR10 of the online test-time fine-tuning. We use the same WideResNet-34-10 as in Table 1a, which is trained with $\ell_\infty$ budget 0.031. The inputs are in the $\ell_\infty$ ball of $\varepsilon = 0.015$. The self-supervised task is the ensemble of rotation and vertical flip.

| Methods | Square Attack | | PGD-20 | | AutoPGD | | FAB | |
|---|---|---|---|---|---|---|---|---|
| | Standard | Adaptive | Standard | Adaptive | Standard | Adaptive | Standard | Adaptive |
| Meta AT w/o FT | 78.01% | - | 75.34% | - | 72.72% | - | 72.58% | - |
| Online FT | 80.50% | 79.87% | 77.14% | 76.75% | 77.25% | **74.93%** | 82.04% | 83.76% |

Table 9: Ablation study on the online test-time fine-tuning. The dataset is CIFAR10 and the task is the "Rotation + VFlip". All attacks are standard attacks. Removing the $\mathcal{L}_{SS}$ or $\mathcal{L}_R$ results in lower robust accuracy than the full method. SA stands for Square Attack.

| Methods | SA | PGD-20 | AutoPGD | FAB |
|---|---|---|---|---|
| Before FT | 65.75% | 59.51% | 53.99% | 53.85% |
| Online FT | **67.34%** | **61.79%** | **59.23%** | **76.39%** |
| Removing $\mathcal{L}_R$ | 66.83% | 60.45% | 57.32% | 75.63% |
| Removing $\mathcal{L}_{SS}$ | 65.44% | 60.24% | 55.64% | 75.08% |

Table 10: Accuracy on clean images. Networks are trained with corresponding meta adversarial training.

| Methods | Rotation | VFlip | Rotation + VFlip |
|---|---|---|---|
| Without FT | 84.77% | 86.55% | 86.36% |
| Online FT | **86.10%** | **87.00%** | **87.10%** |

of test-time adaptation is necessary. Reducing the inference time is an important future work of these kinds of methods.

**Combination with TRADES (Zhang et al., 2019).** Table 13 shows the robust accuracy of combining our test-time adaptation with TRADES. Our test-time adaptation improves the robust accuracy by about 4%, which shows our approach can improve various types of robust training methods.

## C.4 VISUALIZATION

In Figure 4, we show the visualization of several examples that our test-time adaptation successfully corrects the misclassified examples. The input examples are generated by AutoPGD on CIFAR10, and we fine-tune the network with the ensemble of Rotation and VFlip tasks. It shows our test-time adaptation reduces the loss for the whole neighbourhood of the input examples to increase the accuracy of the model.

In Figure 5, we show the histograms of the loss values for the successful and unsuccessful test-time adapted models. For each input instance, if the test-time adaptation corrects the wrong prediction, we count it as successful. And if the misclassified instance is not correctly predicted after our test-time adaptation, it is counted as an unsuccessful one. The figure illustrates that our method can adapt the model to correctly classify the instances close to the decision boundary (with medium loss value). However, for the highly misclassified instances (with large loss value), which are far away from the decision boundary, our test-time adaptation cannot make the model change so much to predict correct labels for them.

## D DETAILS OF OUR EXPERIMENTAL SETTING

### D.1 HYPERPARAMETERS

**Meta Adversarial Training.** The algorithm of Meta Adversarial Training is shown in Algorithm 1. We consider an $\ell_\infty$ norm with an adversarial budget $\varepsilon = 0.031$. We also use two different network architectures: WideResNet-34-10 for CIFAR10 and ResNet18 for STL10 and Tiny ImageNet. Following the common settings for AT, we train the network for 100 epochs using SGD with a

Table 11: Experiments to rule out the possibility of label leaking. We use the WideResNet-34-10 trained with $\ell_\infty$ budget $\varepsilon = 0.031$ and show the robust test accuracy on CIFAR10 of the online test-time fine-tuning. The self-supervised task is the ensemble of rotation and vertical flip.

| Methods | Square Attack | | PGD-20 | | AutoPGD | | FAB | |
|---|---|---|---|---|---|---|---|---|
| | Standard | Adaptive | Standard | Adaptive | Standard | Adaptive | Standard | Adaptive |
| Meta AT w/o FT | 85.43% | - | 85.60% | - | 85.00% | - | 86.22% | - |
| Online FT | 86.56% | 87.63% | 86.29% | 86.68% | 86.10% | 85.90% | 87.61% | 86.97% |

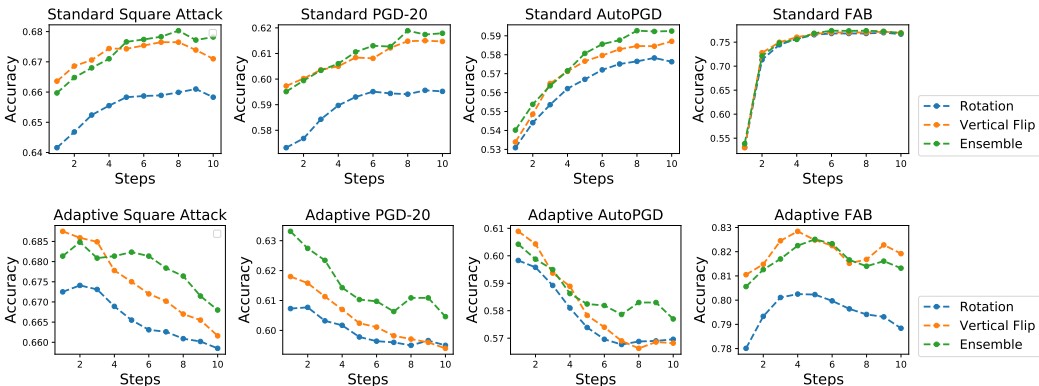

Figure 3: Robust accuracy at different steps of the online test-time fine-tuning on CIFAR10.

Table 12: Average inference time for each instance using different methods.

| Batch Size | 1 | 5 | 10 | 20 | 40 |
|---|---|---|---|---|---|
| Normal | 17.1ms | 14.5ms | 13.2ms | 12.8ms | 11.7ms |
| SOAP (Shi et al., 2020) | 163ms | 91.2ms | 75.3ms | 73.1ms | 72.5ms |
| Ours | 545ms | 168ms | 118ms | 83.9ms | 82.9ms |

momentum factor of 0.9 and a weight decay factor of $5 \times 10^{-4}$. The learning rate $\beta$ starts at 0.1 and is divided by a factor of $10$ after the 50-th and again after the 75-th epochs. The step size $\alpha$ in Eqn (8) is the same as $\beta$. The factor $C'$ in Eqn (11) is set to $1.0$. We use 10-iteration PGD (PGD-10) with a step size of 0.007 to find the adversarial image $B_j^\star$ at training time. The weight of each self-supervised task is set to $C_k = \frac{1}{K}$. We set $|B_j| = 32$ and sample 8 batches $B_1, ..., B_8$ in each iteration. Furthermore, we save the model after the 51-st epoch for further evaluation, as the model obtained right after the first learning rate decay usually yields the best performance (Rice et al., 2020).

We use PGD with the standard cross-entropy loss to generate adversarial examples at training time in line 3, line 6 and line 8 of Algorithm 1. The hyperparameters of the attacks are as follows:

- Line 3: PGD-10 with step size $0.007$.
- Line 6: As $\boldsymbol{\theta}_j^*$ is similar to $\boldsymbol{\theta}$, the adversarial examples at this step are similar to those at Line 4. To save training time, we therefore choose the starting point of the attack as the adversarial examples in Line 4 and use PGD-2 with a step size of $0.005$.
- Line 8: PGD-3 with step size $0.02$.

**Online Test-time Fine-tuning.** The algorithm for online fine-tuning is shown in Algorithm 2. We fine-tune the network for $T = 10$ steps with a momentum of 0.9 and a learning rate of $\eta = 5 \times 10^{-4}$. We set $C_k = \frac{1}{K}$ and $C = 15.0$. In line 2 of Algorithm 2, we sample a batch $B \subset D$ containing 20 training images. In line 3, we use PGD-10 with a step size of 0.007.

**Offline Test-time Fine-tuning.** The algorithm for offline fine-tuning is shown in Algorithm 3. As stochastic gradient descent is more efficient for a large amount of data, we use stochastic gradient descent in the offline fine-tuning. This is the main difference between Algorithm 2 (online fine-

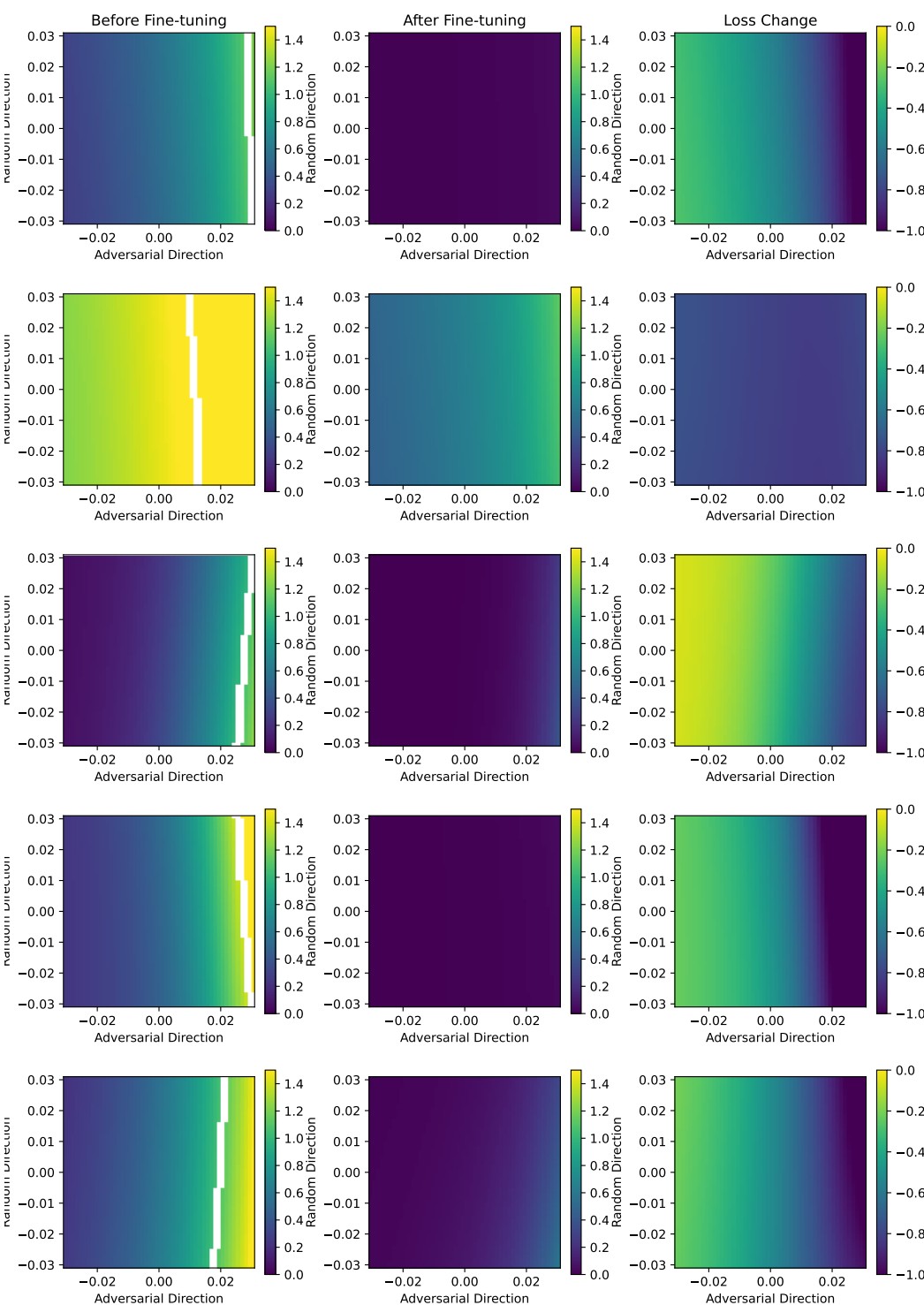

Figure 4: Visualization of several examples that our test-time adaptation successfully changes the wrong prediction. Each row represents an example of loss surfaces before fine-tuning, after fine-tuning and the loss changes of our fine-tuning. The origin point represents the clean example. Following (Kim et al., 2021), $x$-axis represents the direction of the adversarial example and $y$-axis is a random direction. The white line is the decision boundary. As the fine-tuned model correctly classifies the input example, the decision boundary does not exist in the neighbourhood of the clean input for the fine-tuned model.

Table 13: Combination with our test-time adaptation with TRADES on CIFAR10 with the ensemble of rotation and vertical flip tasks.

| | Standard AutoPGD | Adaptive AutoPGD | GMSA AutoPGD |
|---|---|---|---|
| Meta AT w/o FT | 54.06% | | |
| Online FT | 59.63% | 57.74% | 59.39% |

Figure 5: Histograms of the loss values for the successful and unsuccessful test-time adapted models.

tuning) and Algorithm 3 (offline fine-tuning). We also fine-tune the network for 10 epochs. The batch size of each $\widetilde{B}_j^\star$ is 128. The other hyperparameters are the same as in the online version.

---

**Algorithm 2** Self-supervised Test-time Fine-tuning

---

**Input:** Initial parameters $\boldsymbol{\theta}^0$; Adversarial test images $\widetilde{B}^\star = \{\widetilde{\mathbf{x}}_i^\star\}_{i=1}^b$; Training data $D$; Learning rate $\eta$; Steps $T$; Weights $C_k$ and $C$
**Output:** Prediction of $\widetilde{\mathbf{x}}_i^\star$: $\widehat{y}_i$
1: **for** $t = 1$ to $T$ **do**
2:      Sample a batch of training images $B \subset D$
3:      Find adversarial $\mathbf{x}_i^\star$ of training image $\mathbf{x}_i \in B$ by PGD attack.
4:      Calculate $\mathcal{L}_{test}$ in Eqn (6)
5:      $\boldsymbol{\theta}^t = \boldsymbol{\theta}^{t-1} - \eta \nabla_{\boldsymbol{\theta}^{t-1}} \mathcal{L}_{test}(\widetilde{B}^\star, B; \boldsymbol{\theta}^{t-1})$
6: **end for**
7: **return** Prediction $\widehat{y}_i = \arg\max_j F(\widetilde{\mathbf{x}}_i^\star; \boldsymbol{\theta}^T)_j$

---

**Attacks.** The detailed settings of each attack are provided below:

- PGD-20. We use 20 iterations of PGD with step size $\gamma = 0.003$. The attack loss is the cross-entropy.

- AutoPGD. We use both the cross-entropy and the difference of logits ratio (DLR) as the attack loss. The hyperparameters are the same as in (Croce & Hein, 2020a).

- FAB. We use the code from (Croce & Hein, 2020a) and keep the hyperparameters the same.

- Square Attack. We set $T = 2000$ and the initial fraction of the elements $p = 0.3$. The other hyperparameters are the same as in (Andriushchenko et al., 2020).

For the adaptive versions, we set the interval $u = \lceil T/5 \rceil$.

### D.2 SELF-SUPERVISED TASKS

**Rotation Prediction** is a widely used self-supervision task proposed in (Gidaris et al., 2018) and has been employed in AT as an auxiliary task to improve the robust accuracy (Chen et al., 2020a; Hendrycks et al., 2019). Following (Gidaris et al., 2018), we create 4 copies of the input image by rotating it with $\Omega = \{0°, 90°, 180°, 270°\}$. The task then consists of a 4-way classification problem, where the head $g_{\text{rotate}}$ aims to predict the correct rotation angle. The loss for an image $\mathbf{x}$ is the average

---

**Algorithm 3** Self-supervised Test-time Fine-tuning with SGD

---

**Input:**

Initial parameters $\boldsymbol{\theta}^0$; Adversarial test images $\widetilde{B}^\star = \{\widetilde{\mathbf{x}}_i^\star\}_{i=1}^b$; Training data $D$; Learning rate $\eta$; Steps $T$; Weights $C_k$ and $C$

**Output:** Prediction of $\widetilde{\mathbf{x}}_i^\star$: $\hat{y}_i$

1: **for** $t = 1$ to $T$ **do**
2:     Divide $\widetilde{B}^\star$ into $r$ subsets $\widetilde{B}_1^\star, ..., \widetilde{B}_r^\star$
3:     **for** $\widetilde{B}_j^\star$ in $\widetilde{B}_1^\star, ..., \widetilde{B}_r^\star$ **do**
4:         Sample a batch of training images $B \subset D$
5:         Find adversarial $\mathbf{x}_i^\star$ of training image $\mathbf{x}_i \in B$ by PGD attack.
6:         $\boldsymbol{\theta}^t = \boldsymbol{\theta}^{t-1} - \eta \nabla_{\boldsymbol{\theta}^{t-1}} \mathcal{L}_{test}(\widetilde{B}_j^\star, B; \boldsymbol{\theta}^{t-1})$
7:     **end for**
8: **end for**
9: **return** Prediction $\hat{y}_i = \arg\max_j F(\widetilde{\mathbf{x}}_i^\star; \boldsymbol{\theta}^T)_j$

---

cross-entropy over the 4 copies, given by

$$\mathcal{L}_{\text{rotate}}(\mathbf{x}) = -\frac{1}{4} \sum_{\omega \in \Omega} \log(G_{\text{rotate}}(\mathbf{x}_\omega)_\omega) , \qquad (37)$$

where $\mathbf{x}_\omega$ is the rotated image with angle $\omega \in \Omega$, $G_{\text{rotate}} = g_{\text{rotate}} \circ E$ denotes the classifier for rotation prediction, and $G_{\text{rotate}}(\cdot)_\omega$ is the predicted probability for the $\omega$ angle. The head $g_{\text{rotate}}$ is a fully-connected layer followed by a softmax layer.

**Vertical Flip (VFlip) Prediction** is a self-supervised task similar to rotation prediction and has also been used for self-supervised learning (Saito et al., 2020). In essence, we make two copies of the input image and flip one copy vertically. The head $g_{\text{vflip}}$ then contains a 2-way fully-connected layer followed by a softmax layer and predicts whether the image is vertically flipped or not. The corresponding loss for an image $\mathbf{x}$ is

$$\mathcal{L}_{\text{vflip}}(\mathbf{x}) = -\frac{1}{2} \sum_{v \in V} \log(G_{\text{vflip}}(\mathbf{x}_v)_v) , \qquad (38)$$

where $V = \{\text{flipped}, \text{not flipped}\}$ is the operation set and $G_{\text{vflip}} = g_{\text{vflip}} \circ E$. $\mathbf{x}_v$ denotes and transformed input and $G_{\text{vflip}}(\cdot)_v$ is the probability of operation $v$. Note that we do not flip the image horizontally as it is a common data augmentation technique and classifiers typically seek to be invariant to horizontal flip.

## E ADAPTIVE ATTACKS

In the white-box attacks, the attacker knows every detail of the defense method. Therefore, we need to assume that the attacker is aware of our test-time adaptation method and will adjust its strategy for generating adversarial examples accordingly. Here, we discuss one such strong adaptation strategy targeted to our method.

Suppose that the attacker is fully aware of the hyperparameters for test-time adaptation. Then, finding adversaries $\widetilde{B}^\star$ of the clean subset $\widetilde{B}$ can be achieved by maximizing the adaptive loss

$$\widetilde{\mathbf{x}}_i^\star = \arg\max_{\|\widetilde{\mathbf{x}}_i^\star - \mathbf{x}_i\| \leq \varepsilon} \mathcal{L}_{attack}(F(\widetilde{\mathbf{x}}_i^\star), y; \boldsymbol{\theta}^T(\widetilde{B}^\star)) , \qquad (39)$$

where $\mathcal{L}_{attack}$ refers to the general attack loss, such as the cross-entropy or the difference of logit ratio (DLR) (Croce & Hein, 2020a). Let $\boldsymbol{\theta}^T$ be the fine-tuned test-time parameters using Algorithm 2. At the $k$-th step of the attack, it depends on the input $\widetilde{B}^{(k)} = \{(\widetilde{\mathbf{x}}_j^{(k)}, \widetilde{y}_j)\}_{j=1}^b$ via the update

$$\boldsymbol{\theta}^{t+1} = \boldsymbol{\theta}^t - \eta \nabla_{\boldsymbol{\theta}^t} \mathcal{L}_{test}(\widetilde{B}^{(k)}, B) , \qquad (40)$$

where $\mathcal{L}_{test}$ and $B$ are the loss function and subset of training images mentioned in Eqn (6). As $\boldsymbol{\theta}^T$ is a function of the input $\widetilde{B}^{(k)}$, we can calculate the end-to-end gradient of $\widetilde{\mathbf{x}}_i^{(k)} \in \widetilde{B}^{(k)}$ as

$\nabla_{\widetilde{\mathbf{x}}_i^{(k)}}\mathcal{L}_{attack}(F(\widetilde{\mathbf{x}}_i^{(k)});\boldsymbol{\theta}^T(\widetilde{B}^{(k)}))$. However, $\boldsymbol{\theta}^T$ goes through $T$ gradient descent steps, and thus calculating the gradient $\nabla_{\widetilde{\mathbf{x}}_i^{(k)}}\boldsymbol{\theta}^T(\widetilde{B}^{(k)})$ requires $T$-th order derivatives of the backbone $E$, which is virtually impossible if $T$ or the dimension of $\boldsymbol{\theta}_E$ is large. We therefore approximate the gradient as

$$\text{Grad}(\widetilde{\mathbf{x}}_i^{(k)}) \approx \nabla_{\widetilde{\mathbf{x}}_i^{(k)}}\mathcal{L}_{attack}(F(\widetilde{\mathbf{x}}_i^{(k)});\boldsymbol{\theta}^T) \ , \tag{41}$$

which treats $\boldsymbol{\theta}^T$ as a fixed variable so that high-order derivatives from $\boldsymbol{\theta}^T(\widetilde{B}^{(k)}(\widetilde{\mathbf{x}}_i^{(k)}))$ can be avoided. Although this approximation makes $\text{Grad}(\widetilde{\mathbf{x}}_i^{(k)})$ inaccurate, common white-box attacks use projected gradients, which are robust to such inaccuracies. For example, PGD only uses the sign of the gradient under an $\ell_\infty$ adversarial budget. Note that solving the maximization in Eqn (12) does not necessarily require calculating the gradient $\text{Grad}(\widetilde{\mathbf{x}}_i^{(k)})$. For instance, we will also use Square Attack (Andriushchenko et al., 2020), a strong score-based black-box attack, to maximize Eqn (12) and generate adversaries for $\widetilde{B}$.

As another approximation to save time, one can also fix $\boldsymbol{\theta}^T$ for several iterations. This leverages the intuition that attack strategies often make small changes to the input $\widetilde{\mathbf{x}}$, and thus, for the intermediate images in the $k$-th and $(k+1)$-th steps, $\boldsymbol{\theta}^T(\widetilde{B}^{(k)})$ and $\boldsymbol{\theta}^T(\widetilde{B}^{(k+1)})$ should be close. Therefore, a general version of our adaptive attacks only updates $\boldsymbol{\theta}^T$ every $u$ iterations, with $u$ a hyperparameter.

In Algorithm 4, 6, 5 and 7, we show the algorithms for $\ell_\infty$ norm-based adaptive PGD, AutoPGD, Square Attack and FAB, respectively. The main difference between the original and adaptive versions is the target loss function for maximization. The reader may refer to (Andriushchenko et al., 2020; Croce & Hein, 2020a;b) for a more detailed description of the steps in these algorithms (*e.g.*, the condition for decreasing the learning rate in AutoPGD).

---

**Algorithm 4** $\ell_\infty$ Norm Adaptive PGD Attack

---

**Input:** Test images $\widetilde{B} = \{(\widetilde{\mathbf{x}}_i, \widetilde{y}_i)\}$; Attack loss $\mathcal{L}_{attack}$; Step size $\gamma$; Iterations $T$; Intervals $u$; Adversarial budget $\varepsilon$; Trained parameters of the network $\boldsymbol{\theta}^0$.
**Output:** Adversarial images $\widetilde{B}^\star = \{\widetilde{\mathbf{x}}_i^\star\}$
 1: Add random noise to $\widetilde{\mathbf{x}}_i$ in $\widetilde{B}$ and get $\widetilde{B}'$
 2: **for** $t = 1$ to $T$ **do**
 3:     **if** $t \bmod u = 0$ **then**
 4:         Get final parameters $\boldsymbol{\theta}^T$ by taking $\widetilde{B}'$ as input image for Algorithm 2: $\boldsymbol{\theta} = \boldsymbol{\theta}^T$
 5:     **end if**
 6:     **for** $\widetilde{\mathbf{x}}_i'$ in $\widetilde{B}'$ **do**
 7:         $\text{Grad}(\widetilde{\mathbf{x}}_i') = \nabla_{\widetilde{\mathbf{x}}_i'}\mathcal{L}_{attack}(F(\widetilde{\mathbf{x}}_i'), \widetilde{y}_i; \boldsymbol{\theta})$
 8:         $\widetilde{\mathbf{x}}_i' = \text{Clip}_{[\widetilde{\mathbf{x}}_i-\varepsilon, \widetilde{\mathbf{x}}_i+\varepsilon]}(\widetilde{\mathbf{x}}_i' + \gamma\text{Sign}(\text{Grad}(\widetilde{\mathbf{x}}_i')))$
 9:     **end for**
10: **end for**
11: **return** Adversarial image $\widetilde{\mathbf{x}}_i^\star = \widetilde{\mathbf{x}}_i'$

---

---

**Algorithm 5** $\ell_\infty$ Norm Adaptive AutoPGD

---

**Input:** Test images $\widetilde{B} = \{(\widetilde{\mathbf{x}}_i, \widetilde{y}_i)\}$; Attack loss $\mathcal{L}_{attack}$; Step size $\gamma$; Iterations $T$; Intervals $u$; Adversarial budget $\varepsilon$; Parameter of the adversarially-trained network $\boldsymbol{\theta}^0$; Decay iterations $W = \{w_0, ..., w_n\}$; Momentum $\xi$

**Output:** Adversarial image $\widetilde{B}^\star = \{\widetilde{\mathbf{x}}_i^\star\}$

1: Get final parameter $\boldsymbol{\theta}^T$ by taking $\widetilde{B}$ as input image for Algorithm 2.
2: $\boldsymbol{\theta} = \boldsymbol{\theta}^T$
3: **for** $\widetilde{\mathbf{x}}_i$ in $\widetilde{B}$ **do**
4:     $\widetilde{\mathbf{x}}_i^0 = \widetilde{\mathbf{x}}_i$
5:     $\text{Grad}(\widetilde{\mathbf{x}}_i) = \nabla_{\widetilde{\mathbf{x}}_i}\mathcal{L}_{attack}(F(\widetilde{\mathbf{x}}_i), \widetilde{y}_i; \boldsymbol{\theta})$
6:     $\widetilde{\mathbf{x}}_i^1 = \text{Clip}_{[\widetilde{\mathbf{x}}_i - \varepsilon, \widetilde{\mathbf{x}}_i + \varepsilon]}(\widetilde{\mathbf{x}}_i^0 + \gamma\text{Sign}(\text{Grad}(\widetilde{\mathbf{x}}_i)))$
7:     $l_i^0 = \mathcal{L}_{attack}(F(\widetilde{\mathbf{x}}_i^0), \widetilde{y}_i; \boldsymbol{\theta})$
8:     $l_i^1 = \mathcal{L}_{attack}(F(\widetilde{\mathbf{x}}_i^1), \widetilde{y}_i; \boldsymbol{\theta})$
9:     $l_i^* = \max\{l_i^0, l_i^1\}$
10:     $\widetilde{\mathbf{x}}_i^* = \widetilde{\mathbf{x}}_i^0$ **if** $l_i^* = l_i^0$ **else** $\widetilde{\mathbf{x}}_i^* = \widetilde{\mathbf{x}}_i^1$
11: **end for**
12: **for** $t = 1$ to $T - 1$ **do**
13:     **if** $t \bmod u = 0$ **then**
14:         Get final parameter $\boldsymbol{\theta}^T$ by taking $\widetilde{B}^* = \{\widetilde{\mathbf{x}}_i^*\}$ as input image for Algorithm 2.
15:         $\boldsymbol{\theta} = \boldsymbol{\theta}^T$
16:     **end if**
17:     **for** $i = 1, ..., |\widetilde{B}|$ **do**
18:         $\text{Grad}(\widetilde{\mathbf{x}}_i^t) = \nabla_{\widetilde{\mathbf{x}}_i^t}\mathcal{L}_{attack}(F(\widetilde{\mathbf{x}}_i^t), \widetilde{y}_i; \boldsymbol{\theta})$
19:         $z_i^{t+1} = \text{Clip}_{[\widetilde{\mathbf{x}}_i - \varepsilon, \widetilde{\mathbf{x}}_i + \varepsilon]}(\widetilde{\mathbf{x}}_i^t + \gamma\text{Sign}(\text{Grad}(\widetilde{\mathbf{x}}_i^t)))$
20:         $\widetilde{\mathbf{x}}_i^{t+1} = \text{Clip}_{[\widetilde{\mathbf{x}}_i - \varepsilon, \widetilde{\mathbf{x}}_i + \varepsilon]}(\widetilde{\mathbf{x}}_i^t + \xi(z_i^{t+1} - z_i^t) + (1 - \xi)(\widetilde{\mathbf{x}}_i^t - \widetilde{\mathbf{x}}_i^{t-1}))$
21:         $l_i^{t+1} = \mathcal{L}_{attack}(F(\widetilde{\mathbf{x}}_i^{t+1}), \widetilde{y}_i; \boldsymbol{\theta})$
22:         $\widetilde{\mathbf{x}}_i^* = \widetilde{\mathbf{x}}_i^{t+1}$ and $l_i^* = l_i^{t+1}$ **if** $l_i^{t+1} > l_i^*$
23:         **if** $k \in W$ and satisfy the condition of dropping learning rate **then**
24:             $\gamma = \gamma/2$ and $\widetilde{\mathbf{x}}_i^{t+1} = \widetilde{\mathbf{x}}_i^*$
25:         **end if**
26:     **end for**
27: **end for**
28: **return** Adversarial image $\widetilde{\mathbf{x}}_i^* = \widetilde{\mathbf{x}}_i^*$

---

---

**Algorithm 6** $\ell_\infty$ Norm Adaptive Square Attack

---

**Input:** Test images $\widetilde{B} = \{(\widetilde{\mathbf{x}}_i, \widetilde{y}_i)\}$; Attack loss $\mathcal{L}_{attack}$; Step size $\gamma$; Iterations $T$; Intervals $u$; Image size $w$; Color channels $c$; Adversarial budget $\varepsilon$; Parameter of the adversarially-trained network $\boldsymbol{\theta}^0$.
**Output:** Adversarial image $\widetilde{B}^\star = \{\widetilde{\mathbf{x}}_i^\star\}$
 1: Add noise to $\widetilde{\mathbf{x}}_i$ in $\widetilde{B}$ and get $\widetilde{B}'$
 2: **for** $t = 1$ to $T$ **do**
 3:    **if** $t \bmod u = 0$ **then**
 4:       Get final parameter $\boldsymbol{\theta}^T$ by taking $\widetilde{B}'$ as input image for Algorithm 2.
 5:       $\boldsymbol{\theta} = \boldsymbol{\theta}^T$
 6:    **end if**
 7:    **for** $\widetilde{\mathbf{x}}_i'$ in $\widetilde{B}'$ **do**
 8:       $h^t \leftarrow$ side length of the square to modify (according to some schedule)
 9:       $\boldsymbol{\delta} \leftarrow$ array of zeros of size $w \times w \times c$
10:       Sample uniformly $r, s \in \{0, ..., w - h^t\} \subset \mathbb{N}$
11:       **for** $j = 1, ..., c$ **do**
12:          $\rho \leftarrow \text{Uniform}(-2\varepsilon, 2\varepsilon)$
13:          $\boldsymbol{\delta}_{r+1:r+h^t, s+1:s+h^t} = \rho \cdot 1_{h^t \times h^t}$
14:       **end for**
15:       $\widetilde{\mathbf{x}}_i^{\text{new}} = \text{Clip}_{[\widetilde{\mathbf{x}}_i - \varepsilon, \widetilde{\mathbf{x}}_i + \varepsilon]}(\widetilde{\mathbf{x}}_i' + \boldsymbol{\delta})$
16:       $l_i^{\text{new}} = \mathcal{L}_{attack}(F(\widetilde{\mathbf{x}}_i^{\text{new}}), \widetilde{y}_i; \boldsymbol{\theta})$
17:       **if** $l_{\text{new}} < l^*$ **then**
18:          $\widetilde{\mathbf{x}}_i' = \widetilde{\mathbf{x}}_i^{\text{new}}$
19:          $l_i^* = l_i^{\text{new}}$
20:       **end if**
21:    **end for**
22: **end for**
23: **return** Adversarial image $\widetilde{\mathbf{x}}_i^\star = \widetilde{\mathbf{x}}_i'$

---

---

**Algorithm 7** $\ell_\infty$ Norm Adaptive FAB

---

**Input:** Test images $\widetilde{B} = \{(\widetilde{\mathbf{x}}_i, \widetilde{y}_i)\}$; Step size $\gamma$; Iterations $T$; Intervals $u$; Adversarial budget $\varepsilon$; Trained parameters of the network $\boldsymbol{\theta}^0$; $\alpha_{\max}, \eta, \beta$.

**Output:** Adversarial images $\widetilde{B}^\star = \{\widetilde{\mathbf{x}}_i^\star\}$

1: Add random noise to $\widetilde{\mathbf{x}}_i$ in $\widetilde{B}$ and get $\widetilde{B}'$
2: $v = +\infty$
3: **for** $t = 1$ to $T$ **do**
4:     **if** $t \bmod u = 0$ **then**
5:         Get final parameters $\boldsymbol{\theta}^T$ by taking $\widetilde{B}'$ as input image for Algorithm 2: $\boldsymbol{\theta} = \boldsymbol{\theta}^T$
6:     **end if**
7:     **for** $\widetilde{\mathbf{x}}_i'$ in $\widetilde{B}'$ **do**
8:         $\mathrm{Grad}(\widetilde{\mathbf{x}}_i')_l = \nabla_{\widetilde{\mathbf{x}}_i'} F(\widetilde{\mathbf{x}}_i'; \boldsymbol{\theta})_l$
9:         $s = \arg\min_{l \neq y_i} \frac{|F(\widetilde{\mathbf{x}}_i'; \boldsymbol{\theta})_l - F(\widetilde{\mathbf{x}}_i'; \boldsymbol{\theta})_{y_i}|}{\|\mathrm{Grad}(\widetilde{\mathbf{x}}_i')_l - \mathrm{Grad}(\widetilde{\mathbf{x}}_i')_{y_i}\|_1}$
10:        $\boldsymbol{\delta}^t = \mathrm{proj}_\infty(\widetilde{\mathbf{x}}_i', \pi_s, C)$
11:        $\boldsymbol{\delta}_{\mathrm{orig}}^t = \mathrm{proj}_\infty(\widetilde{\mathbf{x}}_i, \pi_s, C)$
12:        $\alpha = \min\left\{ \frac{\|\boldsymbol{\delta}^t\|_\infty}{\|\boldsymbol{\delta}^t\|_\infty + \|\boldsymbol{\delta}_{\mathrm{orig}}^t\|_\infty}, \alpha_{\max} \right\} \in [0, 1]$
13:        $\widetilde{\mathbf{x}}_i' = \mathrm{proj}_C\left( (1 - \alpha)(\widetilde{\mathbf{x}}_i' + \eta\boldsymbol{\delta}^t) + \alpha(\widetilde{\mathbf{x}}_i + \eta\boldsymbol{\delta}_{\mathrm{orig}}^t) \right)$
14:        **if** $\widetilde{\mathbf{x}}_i'$ is not classified as $y_i$ **then**
15:           **if** $\|\widetilde{\mathbf{x}}_i' - \widetilde{\mathbf{x}}_i\|_\infty < v$ **then**
16:             $\widetilde{\mathbf{x}}_i^\star = \widetilde{\mathbf{x}}_i'$
17:             $v = \|\widetilde{\mathbf{x}}_i' - \widetilde{\mathbf{x}}_i\|_\infty$
18:           **end if**
19:           $\widetilde{\mathbf{x}}_i' = (1 - \beta)\widetilde{\mathbf{x}}_i + \beta\widetilde{\mathbf{x}}_i'$
20:        **end if**
21:     **end for**
22: **end for**
23: **return** Adversarial image $\widetilde{\mathbf{x}}_i^\star$

---

