# OpenReview forum: "Test-time Adaptation for Better Adversarial Robustness"
_ICLR.cc/2023/Conference — Submitted to ICLR 2023_

### Official Review · Reviewer_CHte · 2022-10-21

**Confidence:** 4
**Correctness:** 3
**Technical Novelty And Significance:** 3
**Empirical Novelty And Significance:** 3
**Recommendation:** 6

**Clarity, Quality, Novelty And Reproducibility:**

I think this paper is well-written and the proposed method is described clearly. The ideas are novel and the results are significant. It provides enough details for the experiments.

**Strength And Weaknesses:**

I think this paper has the following strengths:

1. The idea of using a meta adversarial training method to find a good starting point for test-time adaptation is novel. The ablation study experiment shows that the meta adversarial training is effective in improving robust accuracy.

2. It theoretically shows that the estimators should be test-time adapted in order to achieve the Bayesian optimal adversarial robustness, even for simple models like linear models and the test-time adaptation largely improves the robustness compared with optimal restricted estimators. The theoretical results are significant, though I don't check the correctness of the proofs carefully.

3. The experiments show that the approach is valid on diverse attack strategies, including an adaptive attack strategy that is fully aware of the test-time adaptation, in both the white-box and black-box attacks.

4. The paper is well-written. The proposed defense method is described clearly. It provides enough details of the adaptive attacks used.

However, I think this paper has the following weaknesses:

1.  It doesn't evaluate the Greedy Model Space Attack (GMSA) proposed in [1], which is designed for the transductive-learning based defenses (or the test-time adaptation defenses). I think it is important to include GMSA in the attack evaluation of the proposed defense.

2. It doesn't report the point-wise worse-case robust accuracy under all the attacks considered. For each test data point x, if any of the attacks considered could find an adversarial example x' that fails the defense, then the robust accuracy on x is 0; otherwise, the robust accuracy on x is 1. If we aggregate it over the entire test set, we can get the point-wise worse-case robust accuracy under all the attacks considered. It would be good to include this metric in the experimental results.


[1] Chen, Jiefeng, et al. "Towards Evaluating the Robustness of Neural Networks Learned by Transduction." International Conference on Learning Representations. 2021.

**Summary Of The Paper:**

This paper shows that under suitable assumptions, the Bayesian optimal robust estimator requires test-time adaptation, and such adaptation can lead to a significant performance boost over standard adversarial training. It then proposes self-supervised test-time fine-tuning on adversarially-trained models to improve their generalization ability. A MAT strategy is introduced to find a good starting point for the self-supervised fine-tuning process. Extensive experiments on CIFAR-10, STL10, and Tiny ImageNet demonstrate that the method consistently improves the robust accuracy under different attack strategies, including strong adaptive attacks where the attacker is aware of the test-time adaptation technique.

**Summary Of The Review:**

Overall, I am positive about this paper. I only have some concerns about the adaptive attack evaluation. It is possible that there are stronger adaptive attacks than the ones considered in the paper. I give a weak rejection for now. If the authors could address my concerns, I am willing to raise my scores.

--------POST REBUTTAL--------

The authors have addressed my concerns. Thus, I raise my score from 5 to 6.

---

### Official Review · Reviewer_SHJC · 2022-10-23

**Confidence:** 3
**Correctness:** 2
**Technical Novelty And Significance:** 3
**Empirical Novelty And Significance:** 2
**Recommendation:** 5

**Clarity, Quality, Novelty And Reproducibility:**

### Clarity
Overall the idea of the paper is clearly stated. The clarity can be further improved if (1) The notation becomes simpler and less dense; (2) Some definitions are pretty ad-hoc. For example, the definition of $L_{SS}$ is not presented until the experiment section that discusses what are the self-learning tasks; and (3) Taking several passes to fix the typo.

### Quality
The theoretical part of the work is sound. The empirical evaluation does not convince me that the proposed method is actually more robust (see the Weakness part in the previous review box).


### Novelty
The proposed method is somewhat novel by adapting test-time fine-tuning to improve adversarial robustness.


### Responsibility
The reproducibility can be improved if the author provides a summary paragraph about where to find descriptions that produce the experiments.



**Strength And Weaknesses:**

### Strength

The paper has a solid motivating theorem that shows why a test-time fine-tuning can improve the robustness. Even though the theorem does not directly help the design of the final algorithm, the theoretical contribution is still important. In the empirical part, the paper uses a standard set of adversaries, e.g. AutoAttack and the improvement of robustness is significant compared to its baselines. Overall, the structure of the paper is easy to follow and I know what to expect when I finish reading one section.


### Weakness

My major concern is the empirical evaluation. I also have some minor concerns on the motivating theorem and the writing. Please see the details below.

**The setup of an empirical adversary may not be strong enough.** Firstly let me restate the setup of the problem. If my read on the paper is correct, given a batch of input $X$, this work proposes to fine-tune the parameter of the model $\theta_0 \rightarrow \theta$ and the update $\Delta \theta = \theta - \theta_0$ is a function of the batch $X$ (and some training batch).

To evaluate the robustness, the paper uses two adversaries: a standard and an adaptive one. I am not able to find the definition of a standard adversary but by reading the description of an adaptive one I think the paper assumes that a standard one targets on a model parameterized with $\theta_0$ while an adaptive one targets $\theta$ (please correct me if I am wrong).

The paper assumes the adaptive one is a strong adversary by fully leveraging the knowledge of the fine-tuning process. My comment for this setup is that:

(1) the adaptive one may be just as strong as a standard adversary who is faced with a model without fine-tuning. This is because the white-box adversary has access to the parameters (i.e. $\theta$) of the model that makes the inference instead of some parameters (i.e.$ \theta_0$) that have nothing to do with the inference;

and (2) an adversary who is smarter should be targeting the fine-tuning process. One example I can think of is that the adversary carefully constructs the test batch sent to your system such that these inputs sit evenly on two sides of the decision boundary and are both less than $\epsilon$ away from the boundary. How would the fine-tuning behave? Will it almost make no update to $\theta_0$ (so the adversary can attack the original model again) or it gives up one half the points so after fine-tuning $\theta$ can be robust on the rest? In practice, the adversary can jointly optimize the noise added to each input. For example, if the adversary only cares about the inference result on $x_i$, it can accompany another input $x_j$, together with $x_i, that targets only on making the fine-tuning doing nothing or worse. I drew a picture [here](https://ibb.co/yXPS71D) for a linear classifier to illustrate the case. Also, the training set used in fine-tuning is also exposed to the attacker and a realistic attacker should take advantage of it. In general, I don't think the current adaptive attacker is adaptive enough to the potentially vulnerable parts in the proposed defense.

Two additional questions regarding the experiment:

1. a strictly stronger attacker should always have lower accuracy than a standard one but in Table 1, some adaptive attacker is even worse than a standard one. For example, on the intersection of the row Probation-OnlineFT and the column Square Attack. Can the authors give some explanation to these results?

2. It seems that there are a lot of hyper-parameters to be tuned. Can the author provide some recommendations of ways to find these parameters in the main body of the paper.

**Soft labels used in Theorem 3.1.**  Will the theorem fail to hold if considering hard labels like [0, 1] instead of soft labels generated from Gaussian noise? I think in a classification setup using a soft label might be okay if well-explained motivation is given.

**Typos.** I found many typos in the paper. I list some examples here:

1. Near Theorem 3.1: “However, for ^ arbitrary ratio c = n/d” (an is missing)

2. Near Theorem 3.1: “we plot the of adversarial risk of three estimators for different adversarial budgets, which clearly shows that adaption can significantly …” (missing “our” or “the” before “adaptation”)

3. Near Theorem 3.2: “The theorem shows that when the given input is the adversarial, the test-time adaptation can still ^ lower the adversarial risk” (“is adversarial”)

4. In Figure 1, it should be $\theta_AB$ not $\theta_BA$ from the caption (or I mis-understand the picture).

5. Near Eq 6, “more efficient for more large amount of data” (larger instead of more large)



**Summary Of The Paper:**

This paper proposes a test-time fine-tuning to boost the adversarial robustness of the classifier. Namely, the new method updates the parameter of the underlying deep network based on self-training (and potentially assuming the network is trained with a meta-learning algorithm). To evaluate the robustness, the paper places the model under a regular white-box adversary and a smarter one aware of the fine-tuning and shows that even with a smarter adversary the fine-tuned model is still more robust.

**Summary Of The Review:**

In summary, I am inclined to reject it at this moment because I am not sure if the proposed method actually produces a new network $\theta$ that is more robust with more obviously vulnerable parts exposed to the adversary. The writing of the paper can be improved as well.

---

> ### Comment · Reviewer_SHJC · 2022-11-21
> **Comment on Authors' Response**
>
> Thank you for the response.
>
> The response has managed to addressed some of my concerns but my main concern on the experiment is not resolved. I realize by saying "white-box", the paper actually assumes a weaker version of the setup where the adversary has limited access to the model's parameters. That is, the adversary only has the access to the fine-tuned parameter instead of the both. In this setup, I believe such adversary has a limited "advantage". Moreover, the authors claim that "It shows that even when the training set is known by the attacker, and the attacker uses this information, our test-time adaptation strategy still increases robust accuracy." without commenting on the margin of the improvement. I am a little skeptical that. why this would happen because the adversary has more information but somehow they just fail to incorporate them into the action. Maybe more explanations are necessary to argue the reason behind this observation.
>
> In conclusion, I remain my initial score at this point. Thank you.

---

> > ### Author Response · Authors · 2022-11-27
> > **Follow-up Response to Reviewer SHJC**
> >
> > Thanks for the response from the reviewer. We believe there are some misunderstandings about the setting of our paper. The white-box attack of our paper is provided full information about the original parameter $\theta_0$, and it can infer the parameter $\theta$ by the adaptive attack. The test-time adaptation was also studied by GMSA [1], where the attacker can infer the parameter $\theta$ to find the optimal adversarial targeted at the fine-tuning process. Our test-time adaptation is able to improve the robust accuracy under various strong attack including non-adaptive attack and GMSA.
> > In addition, our method does not have the phenomenon of obfuscated gradient described in [2], where the black-box attack is worse than the white-box attack.
> >
> > The improvements of our method are listed in Table 2, and we summarize them in the following table, which we believe are not marginal improvements.
> >
> > | Cases   | CIFAR10 Rotation | CIFAR10 VFlip  | CIFAR10 Rotation+VFlip | STL10 Rotation+VFlip | Tiny ImageNet Rotation+VFlip |
> > | ------------------- | -------- | ------ | -------------- | -------------- | -------------- |
> > | Online Worst-case Accuracy Improvments | 3.58%   | 3.00% | 3.45%         |  3.90%  |  1.0% |
> > | Offline Worst-case Accuracy Improvments | 4.17%   | 4.03% | 4.12%         | 7.26%  |  3.4% |
> >
> > [1] Chen Jiefeng et al, Towards Evaluating the Robustness of Neural Networks Learned by Transduction, ICLR 2021
> >
> > [2] Anish Athalye et al., Obfuscated Gradients Give a False Sense of Security: Circumventing Defenses to Adversarial Examples, ICML 2018

---

### Official Review · Reviewer_1EpH · 2022-10-24

**Confidence:** 2
**Correctness:** 3
**Technical Novelty And Significance:** 3
**Empirical Novelty And Significance:** 3
**Recommendation:** 5

**Clarity, Quality, Novelty And Reproducibility:**

**Clarity**

The current manuscript was generally hard to follow. The writing could be improved, in particular for Section 1 and 3.

**Quality**

The experiment is performed on various datasets to verify the effectiveness of their method. I believe the quality can be further improved if the method is compared with other AT methods (e.g., TRADES) and explain “why” the proposed method is beneficial for further improving adversarial robustness.

**Novelty**

While the direction that the paper tackled is relatively under-explored, the technical novelty may be limited; it is generally an extension of existing works in improving the robustness of neural networks against the distributional shift.


**Strength And Weaknesses:**

**Strength:**

* The experiments are performed under various datasets to validate the method.
* The paper tackles an under-explored problem of improving adversarial robustness via additional test-time adaptation.
* The method is easy-to-implement, yet shows a considerable improvement in robust accuracy.
* The paper performs a theoretical study that motivates the method

**Weakness:**

* Although I appreciate that the paper includes a discussion (and results) on adaptive attacks (Eq. 12), but I personally doubt that the attack could indeed faithfully find a true (hidden) adversarial examples: (a) the threat model considered in this paper is quite challenging, in a sense that now the attacker should find a sample that minimize the chance of correct adaptation (with a high-dimensional optimization); (b) empirically, the results from adaptive attacks are sometimes worse than the standard attack, which signals that the adaptive attack can be actually worse sometimes.
* The theoretical analysis in Section 3 yet does not fully justify how the proposed method in Section 4 helps to improve the robustness, e.g., to motivate the use of the self-supervision based test-time training objective. Instead, all the explanation in Section 3 deals with a direct AT-like objective; how can such a self-supervised task replace AT as the surrogate objective? Any intuitions/high-level explanations would be welcome. Moreover, at least, for better clarity to readers, I recommend mentioning Section 5.2 in Section 4 that the gradient of the self-supervised objective empirically shows a highly-correlation to a gradient of the AT objective.
* Given that the main claim of the paper is “AT without adaptation cannot be optimal for the best robustness”, the paper should show the superiority of the method by providing a comparison with state-of-the-art AT methods (e.g., TRADES [Zhang et al., 2019]), not limited to regular AT [Madry et al., 2018].
* The technical novelty may be limited: both the pre-training objective (meta adversarial training) and the adaptation objective in the proposed method are an extension of prior works using adversarial examples.
* The paper is generally hard to follow, and the writing could be improved for better clarity. For instance, in the second paragraph in Section 1, “Theoretically, AT does not achieve the optimal robustness. Under suitable assumptions, the Bayesian optimal robust estimator requires test-test adaptation.” appears without detailed explanation or references. It would be much better if the authors can provide several prior relevant works (if it exists) on investigating why the optimal robustness cannot be achieved only with AT and test-time adaptation is a necessary choice for the robust estimator. Otherwise, the paper may include more insights/intuitions on these arguments, as this is the very first part of the paper.

**Questions:**

* The test-time adaptation objectives include the regularization term, which requires the training set: how does the training batch is selected, and is the proposed method robust to the choice of training mini-batches?
* Can the proposed method be combined with other AT objectives (e.g., TRADES), not limited to the vanilla AT objective [Madry et al., 2018]?

**Minor:**

* Typo: test-test adaptation → test-time adaptation (Section 1, second paragraph)
* Typo: Definition 3.2 to 3.2 → Definition 3.2 to 3.4 (Section 4, Theorem 3.1)
* Typo: more large → larger (Section 4.1, last paragraph)


[Madry et al., 2018] Towards Deep Learning Models Resistant to Adversarial Attacks, ICLR 2018

[Zhang et al., 2019] Theoretically Principled Trade-off between Robustness and Accuracy, ICML 2019


**Summary Of The Paper:**

The paper introduces a test-time adaptation method to improve the adversarial robustness of a neural network by proposing (a) a test-time objective based on existing self-supervised training methods (e.g., RotNet [Gidaris et al., 2018]) and (b) a pre-training objective based on incorporating gradient-based meta-learning (e.g., MAML [Finn et al., 2017]) and adversarial training (AT). Experimental results on CIFAR-10, STL10, and Tiny ImageNet demonstrate the effectiveness of the proposed method in improving adversarial robustness under several adversarial attacks.

[Gidaris et al., 2018] Unsupervised Representation Learning by Predicting Image Rotations, ICLR 2018

[Finn et al., 2017] Model-Agnostic Meta-Learning for Fast Adaptation of Deep Networks, ICML 2017


**Summary Of The Review:**

The paper tackles a challenging and under-explored problem of test-time adaptation for adversarial robustness. However, I slightly feel a lack of technical novelty, and that the overall clarity could be improved. For the evaluation side, I am still not fully convinced that the adaptive attacks considered here would faithfully assess the proposed model, which would affect the correctness of defense evaluation. In these respects, I am currently on a slight negative side.

---

### Official Review · Reviewer_J4z7 · 2022-10-25

**Confidence:** 3
**Correctness:** 3
**Technical Novelty And Significance:** 3
**Empirical Novelty And Significance:** 2
**Recommendation:** 5

**Clarity, Quality, Novelty And Reproducibility:**

The paper is well written and explained well. I would have liked if the authors showed more visualizations.

Regarding novelty, I think the idea of using sample specific adaptation at test-time is new. I did not go over the theory section, so I can't comment much. Experimental results show improvements over AT, but I feel more expeirments could have been done to really show how effective the approach can get.

The authors provided code for reproducibility.

**Strength And Weaknesses:**

Strengths:

The idea of using test time training in the context of adversarial robustness is new. The paper advocates for the need for sample-specific decision boundaries as they claim that it is needed to get Bayesian optimal adversarial robustness. To do this, the paper uses a simple self-supervised learning objective on the test samples.

The objective function used is simple and intuitive. The fact that the test time objective also supports a batch size of 1 is nice.

Results clearly show that using test time training improves the accuracy compared to standard adversarial training. The authors also show results on adaptive attacks, which is nice.


Weaknesses:

I think one of the main weaknesses of the paper is that it doesn’t compare with the SOTA approaches such as TRADES, MART, etc. Comparison with SOTA approaches would be nice. The authors can use some benchmarks like RobustBench to compare their models with the best ones out there.

The other thing that is concerning me is the difficulty in training meta-learning based objective. Training models with meta-learning objectives are not straightforward, so it might be hard to scale this to large datasets like Imagenet.

The next issue is the increase in the inference time. From Figure 3, it looks like more adaptation steps are needed to improve the accuracy. This would mean inference time would rise, which could be a concern.

The method proposed by the authors does not depend on what type of adversarial robustness algorithm used. We can simply replace the AT loss with any type of loss used. So, it would be interesting if this method can improve over the best SOTA results. The authors have one experiment in this regard - comparison with Gowal et al.

One of the things I would have liked to seen in this paper is more visualizations. Why is this approach helping? Does the classifier decision boundary change for samples near the decision surface? For the samples far from the decision boundary, do things change? Such visualizations would help improve the understanding of what is happening.

Another interesting experiment would be to see if the approach improves more if the test distribution has more domain shift. I know that the approach was not proposed with this objective in mind, but it might be an additional benefit the authors would get.

The authors use rotation prediction and vertical flip prediction as the self-supervised objective because it doesn’t have dependence on the batch size which makes sense. But these objectives are quite simple ones, and there are other objectives that could be better too. One example is teacher-student objective for self-supervised learning. I am wondering if these approaches would yield better results.


**Summary Of The Paper:**

This paper proposes test time training for the task of learning adversarially robust models. The idea is to finetune the model for the test samples using a combination of self-supervised loss on the test samples and a memory-based loss on a small subset of training samples. Since the original adversarially trained model might not provide a good initialization for finetuning, a meta-learning based objective is proposed that learns a model that gives best finetuning robustness. Experimental results are shown on some benchmark datasets.


**Summary Of The Review:**

I think the approach is interesting. But I feel the paper lacks experimental rigor. Also, some visualizations could have really helped. I feel if authors spend a bit more time with more rigorous experiments and visualizations, this paper would get strong.

---

### Official Review · Reviewer_wAba · 2022-10-26

**Confidence:** 4
**Correctness:** 4
**Technical Novelty And Significance:** 4
**Empirical Novelty And Significance:** 3
**Recommendation:** 6

**Clarity, Quality, Novelty And Reproducibility:**

+ The paper is well organized and their proposed approach is generally well described, although sometimes dense.
+ The approach appears to be novel and well-motivated.
+ The paper includes information on the method that would make it possible to reproduce the experiments.

**Details Of Ethics Concerns:**

None.

**Strength And Weaknesses:**

Strengths:
+ The key concepts and motivations are described in enough detail to understand the paper.   This approach seems easy to implement and employ in practice.
+ Theoretical explanation for the necessity of the strategy and showing that it improves the robust accuracy of the test data. 1. We show that the estimators should be test-time adapted in order to achieve the Bayesian optimal adversarial robustness, even for simple linear models. And the test-time adaptation largely improves the robustness compared with optimal restricted estimators.
+ Extensive empirical results show the benefit of the method, and ablation studies are provided to characterize the approach.
+ The supplementary material provides additional proofs of theorems, implementation details, adaptive attacks, and experimental results that help support the paper.  Also, since the codes are also provided in supplementary material, there is less concern that the results in this paper would be difficult for a reader to reproduce.
Weaknesses:
- The theoretical analysis in Section 3 is dense and somewhat difficult to read.
- The experimental validation is lacking in some respects. The authors should justify their choice of baselines, datasets, and CNN backbones for experiments. Their proposed method should be also compared with SOA methods in terms of time and memory complexity.  There should be further analysis to assess the impact on the performance of class imbalance and batch size.
- It is unclear why offline FT sometimes yields lower accuracy than online FT with the proposed method.

**Summary Of The Paper:**

This paper is about adversarial training for robust accuracy against adversarial attacks. The authors introduce a new procedure to improve the generalization performance of adversarially-trained networks by using self-supervised test-time ﬁne-tuning. In addition, to determine a good starting point for test-time adaptation, a meta-adversarial training strategy based on the MAML framework is proposed that integrates the test-time adaptation procedure during training. This also strengthens the correlation between the self-supervised and classiﬁcation tasks. This new method is validated for different self-supervised tasks on the CIFAR10, STL10, and Tiny ImageNet datasets tasks, and indicates that their method can improve the accuracy of standard adversarial training under diverse white-box and black-box attack strategies.

**Summary Of The Review:**

Overall this is good quality submission.  The proposed method appears to be novel and well-motivated, although the experimental validation could be improved.

---

### Decision · Program_Chairs · 2023-01-20

**Decision:**

Reject

**Justification For Why Not Higher Score:**

The paper can be improved on the experimental rigor, writing and visualizations.

Reviewers are still not fully convinced with the adaptive attack experiments. Specifically, it is still open whether there are indeed any stronger attacks than those considered in the paper. Although the rebuttal incorporates a new recent attack of GSMA, this should be considered more extensively in the experiments (rather than as the ones in Appendix as in the current manuscript). There are two versions of GMSA in the GMSA paper: GMSA-AVG and GMSA-MIN (which one is stronger depends on the defense). In the author response, the authors only implemented GMSA-AVG and gave limited results for this adapted attack (e.g. only evaluate the models' robustness under GMSA-AVG on the CIFAR10 dataset and in the online fine-tuning setting). The authors are suggested to perform more sophisticated evaluations under GMSA attacks and to provide enough details of the GMSA implementation (e.g. details of the GMSA attack process).

In the rebuttal, it was clarified that the GSMA attack performed worse than their initial baseline attack. Given that GSMA is the current state-of-the-art attack method for test-time adaptation based defenses, the paper could have considered additional experiments to give a conclusion on whether (i) there exist a straightforward variant of GSMA that works on the self-supervised learning based defense the paper proposes, or (ii) it is evident enough (either empirically or theoretically) that GSMA is indeed worse and the paper is proposing a new state-of-the-art adaptive attack in their threat model.

The presentation of the paper needs improvement for any potential follow-up works to break/enhance the work. Although the committee appreciates the discussions to bring GMSA evaluation to the paper, reviewers find the technical part of the paper hard to follow. In particular, the reviewers have questions on whether the so-called "adaptive attack" adapts to the defense. Some reviewer's question on why on some entries the adaptive attack shows better robustness is not fully addressed. The readers may need more intuitions/explanations to be convinced that the attack works. Some reviewer is concerned if "adaptive attack" smartly leverages the training batch and other resources to find the adversarial example (as detailed in a thought experiment in one of the initial reviews).

The method proposed in the paper leads to increased computation both in training and inference.  It is unclear whether the empirical gains from the extra computation burden is worth it. The committee suggests adding more visualizations to understand what the test-time training does. This may provide more insights into adversarial robustness.

**Justification For Why Not Lower Score:**

N/A

**Metareview: Summary, Strengths And Weaknesses:**

The paper considers the adversarial robustness problem. The authors made an observation that Bayesian optimal robust estimator requires test-time adaptation, and such adaptation can lead to significant performance boost over standard adversarial training, under suitable assumptions. Motivated by this observation, the authors propose to add a self-supervised test-time adaptation step, to improve the generalization performance of adversarially-trained networks.  To find a good initialization point for test-time adaptation, a meta adversarial training method, which incorporate this test-time adaptation to training,  is proposed.

Strengths:
Designing an adaptive attack for the current threat model is extremely challenging, due to the fundamental difficulty of finding worst-case perturbations of test-time adapted models. The efforts made in the paper can be a useful addition to tackle the challenge in the future.

Weaknesses:
The paper can be improved on the experimental rigor, writing and visualizations.

Reviewers are still not fully convinced with the adaptive attack experiments. Specifically, it is still open whether there are indeed any stronger attacks than those considered in the paper. Although the rebuttal incorporates a new recent attack of GSMA, this should be considered more extensively in the experiments (rather than as the ones in Appendix as in the current manuscript). There are two versions of GMSA in the GMSA paper: GMSA-AVG and GMSA-MIN (which one is stronger depends on the defense). In the author response, the authors only implemented GMSA-AVG and gave limited results for this adapted attack (e.g. only evaluate the models' robustness under GMSA-AVG on the CIFAR10 dataset and in the online fine-tuning setting). The authors are suggested to perform more sophisticated evaluations under GMSA attacks and to provide enough details of the GMSA implementation (e.g. details of the GMSA attack process).

In the rebuttal, it was clarified that the GSMA attack performed worse than their initial baseline attack. Given that GSMA is the current state-of-the-art attack method for test-time adaptation based defenses, the paper could have considered additional experiments to give a conclusion on whether (i) there exist a straightforward variant of GSMA that works on the self-supervised learning based defense the paper proposes, or (ii) it is evident enough (either empirically or theoretically) that GSMA is indeed worse and the paper is proposing a new state-of-the-art adaptive attack in their threat model.

The presentation of the paper needs improvement for any potential follow-up works to break/enhance the work. Although the committee appreciates the discussions to bring GMSA evaluation to the paper, reviewers find the technical part of the paper hard to follow. In particular, the reviewers have questions on whether the so-called "adaptive attack" adapts to the defense. Some reviewer's question on why on some entries the adaptive attack shows better robustness is not fully addressed. The readers may need more intuitions/explanations to be convinced that the attack works. Some reviewer is concerned if "adaptive attack" smartly leverages the training batch and other resources to find the adversarial example (as detailed in a thought experiment in one of the initial reviews).

The method proposed in the paper leads to increased computation both in training and inference.  It is unclear whether the empirical gains from the extra computation burden is worth it. The committee suggests adding more visualizations to understand what the test-time training does. This may provide more insights into adversarial robustness.